# JavisDiT++: Unified Modeling and Optimization for Joint Audio-Video Generation

**Kai Liu**[1]*, **Yanhao Zheng**[1]*, **Kai Wang**[3]*, **Shengqiong Wu**[2], **Rongjunchen Zhang**[4],
**Jiebo Luo**[5], **Dimitrios Hatzinakos**[3], **Ziwei Liu**[6], **Hao Fei**[2]†, **Tat-Seng Chua**[2]

[1]Zhejiang University, [2]National University of Singapore, [3]University of Toronto,
[4]HiThink Research, [5]University of Rochester, [6]Nanyang Technological University

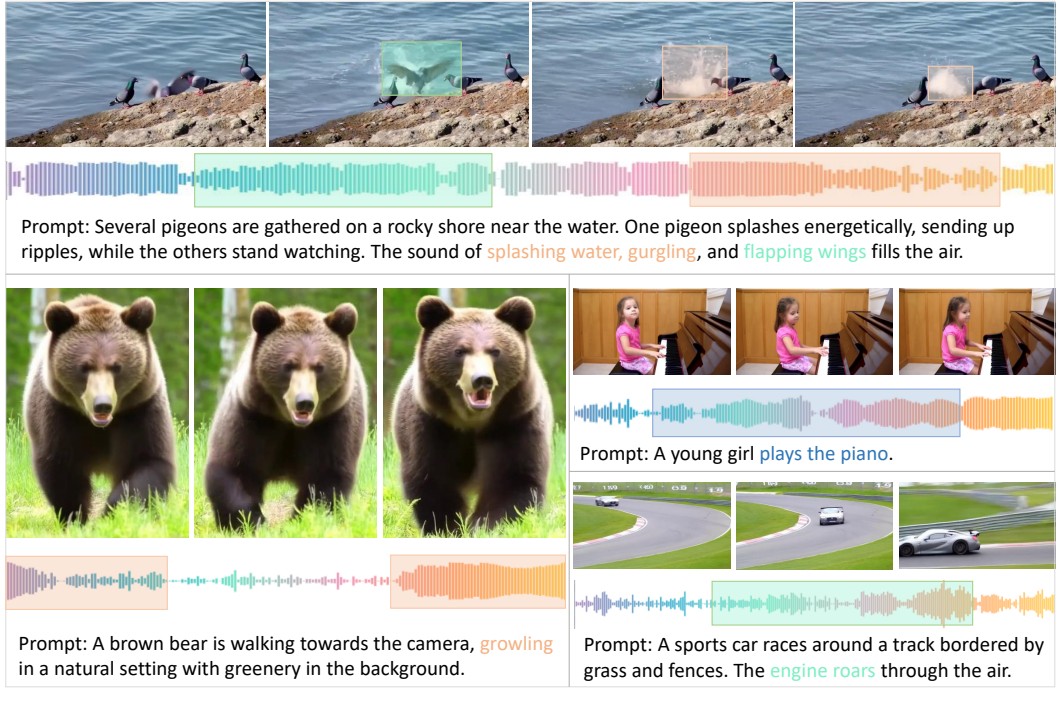

Figure 1: Realistic and diversified joint audio-video generation examples by our JavisDiT++ model.

## ABSTRACT

AIGC has rapidly expanded from text-to-image generation toward high-quality multimodal synthesis across video and audio. Within this context, joint audio-video generation (JAVG) has emerged as a fundamental task that produces synchronized and semantically aligned sound and vision from textual descriptions. However, compared with advanced commercial models such as Veo3, existing open-source methods still suffer from limitations in generation quality, temporal synchrony, and alignment with human preferences. To bridge the gap, this paper presents **JavisDiT++**, a concise yet powerful framework for unified modeling and optimization of JAVG. First, we introduce a modality-specific mixture-of-experts (MS-MoE) design that enables cross-modal interaction efficacy while enhancing single-modal generation quality. Then, we propose a temporal-aligned RoPE (TA-RoPE) strategy to achieve explicit, frame-level synchronization between audio and video tokens. Besides, we develop an audio-video direct preference optimization (AV-DPO) method to align model outputs with human preference across quality,

*Equal contribution. Part of this work was done at HiThink Research. Email: kail@zju.edu.cn
†Corresponding author. Email: haofei7419@gmail.com

consistency, and synchrony dimensions. Built upon Wan2.1-1.3B-T2V, our model achieves state-of-the-art performance merely with around 1M public training entries, significantly outperforming prior approaches in both qualitative and quantitative evaluations. Comprehensive ablation studies have been conducted to validate the effectiveness of our proposed modules. All the code, model, and dataset are released at `https://JavisVerse.github.io/JavisDiT2-page`.

# 1 INTRODUCTION

AI-Generated Content (AIGC) has evolved from text-to-image generation towards more complex domains such as video and audio (Wan et al., 2025; Zheng et al., 2024; Liu et al., 2024b; Jiang et al., 2025). The intrinsic correlation between audio and video modalities has also been increasingly explored, giving rise to related research on video-to-audio (Xing et al., 2024; Cheng et al., 2025; Tian et al., 2025) and audio-to-video (Lyu et al., 2024; Yariv et al., 2024b; Gao et al., 2025b) generation. With the rapidly growing demands for AIGC domains like short videos, film, gaming, and VR, joint audio-video generation (JAVG) from textual inputs has become increasingly crucial in this era (Wang et al., 2025a; Lyu et al., 2025).

MM-Diffusion (Ruan et al., 2023) represents a series of early works that study unconditional audio-video generation, with a scope limited to natural landscapes (Landscape (Lee et al., 2022)) and human dancing (AIST++ (Li et al., 2021)). Then, Uniform (Zhao et al., 2025) extended the task to label-name-conditioned JAVG and evaluated the model on the relatively broader VGGSound dataset (Chen et al., 2020). More recently, JavisDiT (Liu et al., 2025c) and Universe-1 (Wang et al., 2025a) have begun to employ free-form text prompts as inputs, systematically exploring sounding video generation in real-world scenarios. Despite these efforts, current approaches still struggle to produce high-quality, temporally synchronized sounding videos compared with advanced proprietary models such as Veo3 (DeepMind, 2025). Fig. 2 quantitatively shows the gap between previous methods and Veo3 (details are provided in Sec. B.3), indicating current JAVG methods fail to capture human preferences regarding the aesthetics and harmony of audio-visual content.

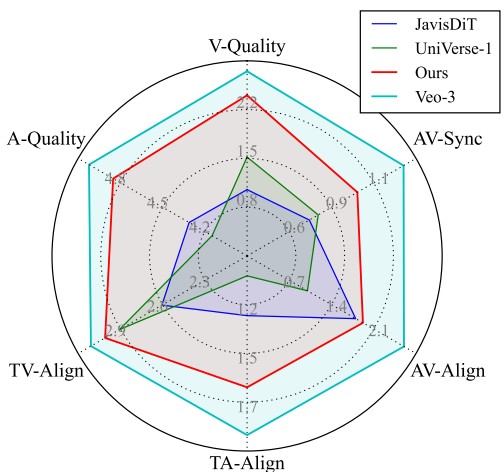

Figure 2: Comparison with recent JAVG models.

To bridge the gap, we provide a unified perspective of JAVG modeling and optimization to generate *high-quality*, *better-synchronized*, and *human-preference-aligned* sounding videos.

First, to improve generation quality, we introduce a **modality-specific mixture-of-experts (MS-MoE)** module, where audio and video tokens exchange information through shared multi-head self-attention layers, and then aggregate intra-modal information via two separate FFN layers (Deng et al., 2025). This architecture enhances single-modality generation quality compared with Uniform (Zhao et al., 2025), which processes aggregated audio-video tokens using a single FFN module. On the other hand, MS-MoE is simpler, more efficient, and more unified than dual-DiT plus audio-visual interaction block designs such as JavisDiT (Liu et al., 2025c) and Universe-1 (Wang et al., 2025a). Sec. C provides detailed architectural comparison with related works.

Second, to enhance audio-video synchrony, we propose an **temporal-aligned multimodal RoPE (TA-RoPE)** strategy, which aligns the position IDs of audio and video tokens on a unified temporal axis (Xu et al., 2025), enabling direct and frame-level fine-grained temporal synchronization. On one hand, this strategy modulates audio-video temporal synchrony more explicitly and effectively against the ST-Prior in JavisDiT (Liu et al., 2025c) and the Stitching strategy in Universe-1 (Wang et al., 2025a). On the other hand, TA-RoPE is actually compatible with these methods — it can be combined with ST-Prior and frame-level cross-attention for slightly better performance. However, we ultimately abandon these combinations to maintain overall simplicity and efficiency.

Finally, to capture human preference, we design an **audio-video direct preference optimization (AV-DPO)** method to align the model with collected preference data (Liu et al., 2025b), further enhancing video-audio generation quality and synchronization. To curate robust preference data, we leverage diverse reward models to comprehensively evaluate the generated audio and video across multiple dimensions (*e.g.*, quality, consistency, and synchrony) and then adopt normalized modality-aware ranking to select winning-losing pairs. To the best of our knowledge, we are the first to introduce preference alignment into JAVG, enabling the model to generate high-quality, synchronized sounding videos, more faithfully aligned with the input text and human preferences.

With the above techniques, our **JavisDiT++** model is built upon Wan2.1-1.3B-T2V (Wan et al., 2025), and is efficiently trained using only 780K diversified audio-text pairs (Liu et al., 2025c) and 360K high-quality sounding videos (Mao et al., 2024), supporting arbitrary duration and resolution ranging from 2–5 seconds and 240p–480p resolution across different aspect ratios. It achieves state-of-the-art performance, surpassing JavisDiT (Liu et al., 2025c) and Universe-1 (Wang et al., 2025a) across both qualitative and quantitative evaluations on various dimensions. Extensive ablation studies demonstrate the effectiveness and rationality of the proposed modules. We hope this work will set a milestone for the field of native joint audio-video generation and provide further inspiration in the field.

In summary, our main contributions are threefold:

- We propose a concise JAVG model architecture, which employs a modality-specific MoE strategy for efficient and high-quality audio-video generation, and introduces temporally aligned RoPE to achieve precise temporal synchronization.

- We are the first to introduce human preference alignment into JAVG, with the AV-DPO algorithm to consistently improve quality, consistency, and synchrony of sounding video generation.

- We train a state-of-the-art JAVG model using only 1M public data entries, hoping to provide a milestone with new inspirations for the field of native joint audio-video generation.

## 2 RELATED WORK

**Joint Audio-Video Generation.** Recent JAVG approaches have taken several forms. Some studies use a unified representation, projecting both modalities into a shared latent space, such as CoDi (Tang et al., 2023; 2024), MM-LDM (Sun et al., 2024), and UniForm (Zhao et al., 2025). However, this strong constraint can cause modality-specific information loss and provide insufficient temporal control. Another line of studies performs intermediate fusion. MM-Diffusion (Ruan et al., 2023), SyncFlow (Liu et al., 2024a), and AV-DiT (Wang et al., 2024) exchange information between modalities within the model's layers using cross-modal attention or adapters. Other methods like Seeing (Xing et al., 2024) and MMDisCo (Hayakawa et al., 2024) use online discriminators to adjust the outputs of separately pre-trained models. More recently, JavisDiT (Liu et al., 2025c) develops a two-stream DiT with a spatio-temporal prior estimator to guide alignment, and UniVerse-1 (Wang et al., 2025a) leverages two pretrained DiTs with a stitching strategy for crossmodal information exchange. However, those methods (Guan et al., 2025; Hu et al., 2025; Low et al., 2025; HaCohen et al., 2026) all rely on complicated (symmetrical or asymmetrical) model architectures and ad-hoc implicit synchrony modulation, hindering the scalability of unified sounding video generation.

**RL in Generative Models.** Reinforcement learning (RL) has been widely applied to align generative models with human preferences. Early approaches use policy-based algorithms such as Proximal Policy Optimization (PPO) to improve diffusion models (Black et al., 2023; Fan et al., 2023), and Direct Preference Optimization (DPO) is subsequently introduced to align text-to-image generation without explicit reward models (Wallace et al., 2024; Yang et al., 2024). This area continues to develop, with recent work improving DPO (Dang et al., 2025; Huang et al., 2025a) and adopting newer strategies like Group-wise Ranking Preference Optimization (GRPO) (Wang et al., 2025b; Xue et al., 2025; Liu et al., 2025a; Yuan et al., 2025). Although these alignment techniques are now widely used for other modalities like video and audio (Zhang et al., 2023; Liu et al., 2025b;d; Furuta et al., 2024; Wu et al., 2025; Gao et al., 2025a; Chen et al., 2025), their application to complex, cross-modal tasks remains limited. To the best of our knowledge, we are the first to successfully apply a preference alignment algorithm to the field of joint audio-video generation.

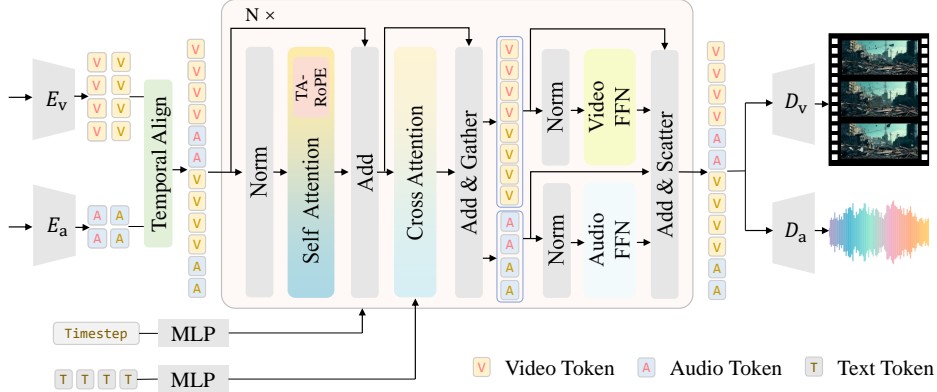

Figure 3: Architecture of JavisDiT++. We use shared attention layers to encourage audio-visual mutual information modeling, with modality-specific FFN layers to enhance intra-modal aggregation. The Temporal-Aligned RoPE strategy is applied to ensure audio-video synchrony. The audio/video embedder layer and prediction head that bridge DiT and VAEs are hidden for simplicity.

## 3 METHODOLOGY

### 3.1 PRELIMINARY

**Flow Matching**. Let $p_0$ denote the target data distribution and $p_1$ be a simple prior distribution (*e.g.*, a standard Guassian $\mathcal{N}(\mathbf{0}, \mathbf{I})$). Rectified Flow (Liu et al., 2023) constructs straight-line paths from noise samples $\mathbf{x}_1 \sim p_1$ to data samples $\mathbf{x}_0 \sim p_0$, where the trajectory $\mathbf{x}_t$ at time $t \in [0, 1]$ is defined as $\mathbf{x}_t = (1 - t)\mathbf{x}_0 + t\mathbf{x}_1$. The corresponding target velocity field becomes $\mathbf{v} = \mathbf{x}_1 - \mathbf{x}_0$. A neural network $\mathbf{v}_\theta(\mathbf{x}_t, t)$ is then trained to regress this target field by minimizing the following objective:

$$\mathcal{L}_{\text{FM}}(\theta) = \mathbb{E}_{t\sim[0,1],\mathbf{x}_0\sim p_0,\mathbf{x}_1\sim p_1} \left[||\mathbf{v}_\theta(\mathbf{x}_t, t) - \mathbf{v}||^2\right] \quad (1)$$

Once trained, new samples can be generated by solving the ordinary differential equation $\mathrm{d}\mathbf{x}/\mathrm{d}t = v_\theta(\mathbf{x}, t)$ from $t = 0$ to $t = 1$, starting from an initial noise sample $\mathbf{x}_0 \sim p_0$.

**Joint Audio-Video Generation**. The goal of JAVG is to model the conditional distribution $p(A, V \mid c)$, where $V \in \mathbb{R}^{T_v \times H \times W \times 3}$ denotes a video with $T_v$ frames of resolution $H \times W$, and $A \in \mathbb{R}^{T_a \times M}$ denotes the audio in a mel-spectrogram with $T_a$ temporal steps and $M$ frequency bins. Given a condition $c$ (*e.g.*, a text prompt), a model $p_\theta$ is trained to generate synchronized audio-video pairs:

$$(\hat{\mathbf{x}}^a, \hat{\mathbf{x}}^v) = \mathbf{v}_\theta(\mathbf{x}_t^a, \mathbf{x}_t^v, t, c); \quad \mathcal{L}_{\text{FM}}^{av}(\theta) = \mathbb{E}_{t\sim[0,1],\mathbf{x}_0\sim p_0,\mathbf{x}_1\sim p_1} \left[||\hat{\mathbf{x}}^a - \mathbf{v}^a||^2 + ||\hat{\mathbf{x}}^v - \mathbf{v}^v||^2\right] \quad (2)$$

**Rotary Position Embedding (RoPE)**. In video generation (Wan et al., 2025), position IDs are assigned along temporal ($T$), height ($H$), and width ($W$) dimensions, and their rotational position embeddings are applied to queries and keys in attention layers:

$$R(t, h, w) = [R_T(t); R_H(h); R_W(w)]; \quad (q_i', k_j') = (R(t_i, h_i, w_i)q_i, R(t_j, h_j, w_j)k_j). \quad (3)$$

This design captures the relative positional relationships of tokens across all three dimensions.

### 3.2 THE DIT MODEL ARCHITECTURE

**From Dual-Stream DiT to a Unified Backbone.** Unlike dual-stream DiT frameworks such as JavisDiT (Liu et al., 2025c) and UniVerse-1 (Wang et al., 2025a), this paper aims to design a *concise*, *efficient*, and *unified* DiT architecture to jointly process audio and video tokens, ensuring better scalability. As shown in Fig. 3, we first flatten and concatenate video and audio tokens, feeding them into subsequent modules for full self-attention to enable dense and rich cross-modal interaction. The tokens are then separated and passed through modality-specific FFNs, which ensure sufficient

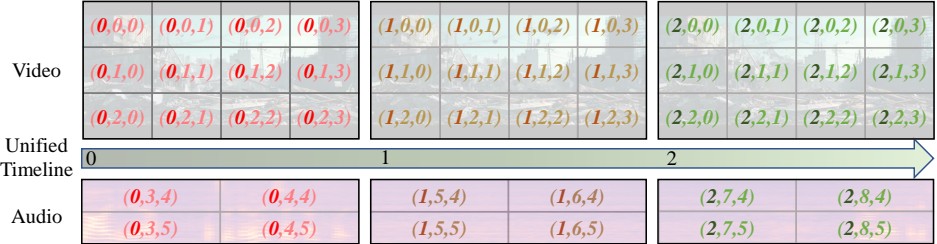

Figure 4: Illustration of temporal-aligned rotary position encoding for video and audio tokens.

intra-modal aggregation. The video VAE from Wan2.1 (Wan et al., 2025) and audio VAE from AudioLDM2 (Liu et al., 2024b) are retained and frozen during the whole training process.

**Modality-Specific Feed-Forward Network.** In contrast to conventional MoE architectures with dynamic routing (Cai et al., 2025), our MS-FFN (or MS-MoE) deterministically assigns audio and video tokens to their modality-specific FFNs/Experts. This design is similar to BAGEL (Deng et al., 2025), which allocates understanding and generation tokens to separate FFNs, while we instead assign tokens based on modality. The advantage is that, after sufficient cross-modal interaction through attention, modality interference in FFN is isolated, allowing each branch to focus on intra-modal feature modeling. Similar to traditional MoE benefits, although the total parameter size increases from 1.3B to 2.1B, the number of activated parameters per token remains 1.3B. Thus, we expand model capacity to improve performance without adding inference overhead.

### 3.3 TEMPORAL-ALIGNED ROTARY POSITION ENCODING

Establishing a shared temporal reference for audio and video tokens is crucial for achieving generation synchrony (Liu et al., 2025c; Wang et al., 2025c). In contrast to the spatial-temporal prior (ST-Prior) employed in JavisDiT (Liu et al., 2025c) and the frame-level cross-attention mechanism in UniVerse-1 (Wang et al., 2025a), this paper proposes a more direct and precise control strategy: an audio-visual Temporally Aligned Rotary Position Encoding (TA-RoPE). Specifically, absolute temporal alignment is enforced along the first dimension (dimension 0) of the 3D position IDs for both audio and video tokens, as illustrated in Fig. 4.

We first retain the 3D RoPE formulation for video tokens as introduced in Wan2.1 (Wan et al., 2025). Given video tokens of shape $T_v \times H \times W$, their 3D position IDs range from $(0,0,0)$ to $(T_v - 1, H - 1, W - 1)$. For audio tokens of shape $T_a \times M$ (extracted from mel-spectrograms as in AudioLDM2 (Liu et al., 2024b)), we first augment them with an additional leading dimension (dimension 0) to align with the video tokens along the absolute time axis. This means, audio tokens corresponding to the same time window of video tokens with temporal ID $i$ are assigned temporal ID $i$ as well. Subsequently, to ensure that audio and video position IDs remain strictly non-overlapping, we offset the original mel-spectrogram dimensions ($T_a$ and $M$) by adding $H$ and $W$, respectively. Consequently, the audio position IDs span from $(H, W)$ to $(H + T_a - 1, W + M - 1)$. Further discussion and comparisons are provided in the Sec. C.

Formally, for an audio token at timestamp $t$ and frequency bin $m$, its positional ID is defined as:

$$R_a(t, m) = \left( \left[ t \cdot \frac{T_v}{T_a} \right], t + H, m + W \right) \tag{4}$$

where $[\cdot]$ denotes the round operation. This formulation explicitly enforces temporal alignment and synchrony between audio and video tokens.

Note that, thanks to the full attention design in Wan2.1, we can logically emulate the interleaved audio-video temporal arrangement shown in Fig. 3 purely through position ID manipulation, without physically reordering tokens. In contrast, within a causal (autoregressive) DiT framework, achieving temporal alignment would require physically interleaving audio and video tokens such that those with smaller temporal position IDs are placed earlier in the sequence. However, such physical reordering entails non-contiguous memory accesses and incurs additional computational overhead.

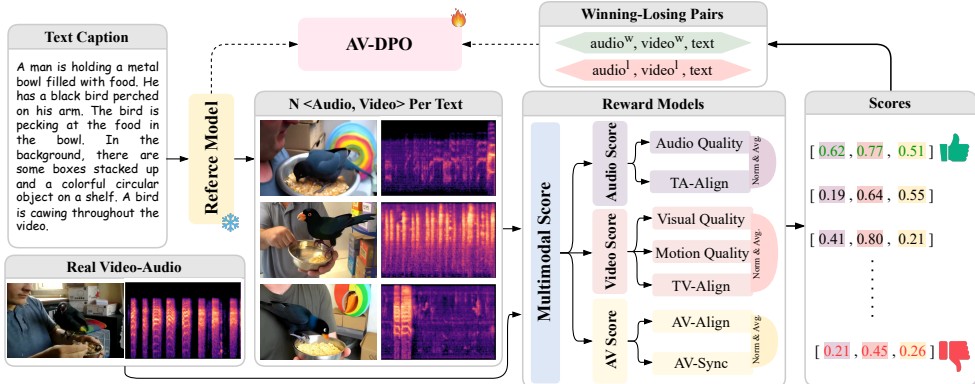

Figure 5: Illustration of preference data collection and training pipeline of audio-video DPO.

## 3.4 DIRECT PERFORMANCE OPTIMIZATION FOR JOINT AUDIO-VIDEO GENERATION

To further improve the video and audio quality and synchronization, we propose an audio-video direct preference optimization (AV-DPO) algorithm to align JAVG models with human preferences. The overall pipeline is shown in Fig. 5, and our designed AV-DPO features some core contributions: (1) AV-DPO is the first one to align rectified flows with preference data for joint audio-video generation; (2) AV-DPO adopts various reward models to automatically and comprehensively evaluate the generated audio-video samples, and performs ranking based on modality-aware dimensions to ensure modality consistency in selected chosen-rejection pairs. The details are introduced as follows.

**Reward Models.** We employ various reward models to evaluate audio-video data from three modality-aware dimensions: audio reward, video reward, and audio-video alignment. Specifically, for audio reward, we leverage AudioBox (Tjandra et al., 2025) to comprehensively evaluate the audio quality, with ImageBind (Girdhar et al., 2023) to measure text-audio semantic similarity (alignment). As for video reward, we take VideoAlign (Liu et al., 2025b) to assess visual and motion quality, with ImageBind (Girdhar et al., 2023) to measure text-video semantic alignment. For audio-video alignment, we also utilize ImageBind to calculate audio-video semantic similarity, with Syncformer (Iashin et al., 2024) to estimate temporal synchrony. To solve the problem that different reward indicators yield various magnitudes, we first normalize the scores of each metric and average them to obtain the reward for the corresponding modality-aware dimension.

**Preference Data Acquisition.** To construct the preference data, we first curate a prompt pool $P_t$ with $30k$ text captions apart from the SFT training data, and then prompt the reference model to generate $N = 3$ audio-video pairs for each prompt. To stabilize the preference optimization, we also add the ground truth audio-video pairs into the generated ones to form the candidate preference data, which are evaluated by the previously mentioned reward models from three modality-specific dimensions. Afterwards, we select the winner sample $a_i^w, v_i^w$ yielding better ranking scores than the loser sample $a_i^l, v_i^l$ across all dimensions:

$$\mathcal{D}_\sqcup = \left\{ (a_i^w, v_i^w, a_i^l, v_i^l, y_i) \mid y_i \in P_t, \mathbb{I}_{u \in \{a,v,av\}} \left\{ \sum_{j=1}^{d_u} \bar{\mathcal{S}}_j^u(a_i^w, v_i^w) > \sum_{j=1}^{d_u} \bar{\mathcal{S}}_j^u(a_i^l, v_i^l) \right\} \right\} \quad (5)$$

where $\bar{\mathcal{S}}_j^u(*)$ represent the normalized score of the $j$-th reward of modality $u$. Finally, eq. (5) results in around $25k$ audio-video preference pairs. It is worth mentioning that winning pairs from our generated samples occupy around $30\%$ of the total preference data, indicating that the baseline model itself already possesses fairly strong generative capabilities.

**Direct Preference Optimization.** Unlike prior single-modality DPO approaches (Liu et al., 2025b), our AV-DPO enhances joint audio-video generation by considering modality-aware preference:

Table 1: Main results on JavisBench for generating 240p4s sounding videos. The best results are marked with **bold**, and the second ones are marked with underline.

| Model | Size | AV-Quality | | Text-Consistency | | | | AV-Consistency | | AV-Synchrony | | Runtime ↓ |
| | | FVD ↓ | FAD ↓ | TV-IB ↑ | TA-IB ↑ | CLIP ↑ | CLAP ↑ | AV-IB ↑ | AVHScore ↑ | JavisScore ↑ | DeSync ↓ | |
| *- T2A+A2V* | | | | | | | | | | | | |
| TempoTkn | 1.3B | 539.8 | - | 0.084 | - | 0.205 | - | 0.139 | 0.122 | 0.103 | 1.532 | 20s |
| TPoS | 1.0B | 839.7 | - | 0.201 | - | 0.229 | - | 0.124 | 0.129 | 0.095 | 1.493 | 19s |
| *- T2V+V2A* | | | | | | | | | | | | |
| ReWaS | 0.6B | - | 9.4 | - | 0.123 | - | 0.280 | 0.110 | 0.104 | 0.079 | 1.071 | 17s |
| See&Hear | 0.4B | - | 7.6 | - | 0.129 | - | 0.263 | 0.160 | 0.143 | 0.112 | 1.099 | 25s |
| FoleyC | 1.2B | - | 9.1 | - | 0.149 | - | 0.383 | 0.193 | **0.186** | 0.151 | 0.952 | 16s |
| MMAudio | 0.1B | - | 6.1 | - | 0.160 | - | 0.407 | **0.198** | 0.182 | 0.150 | 0.849 | 15s |
| *- T2AV* | | | | | | | | | | | | |
| MM-Diff | 0.4B | 2311.9 | 27.5 | 0.080 | 0.014 | 0.181 | 0.079 | 0.119 | 0.109 | 0.070 | 0.875 | 9s |
| JavisDiT | 3.1B | 204.1 | 7.2 | 0.263 | 0.143 | 0.302 | 0.391 | 0.197 | 0.179 | 0.154 | 1.039 | 30s |
| UniVerse-1 | 6.4B | 194.2 | 8.7 | 0.272 | 0.111 | 0.309 | 0.245 | 0.104 | 0.098 | 0.077 | 0.929 | 13s |
| **Ours** | 2.1B | **141.5** | **5.5** | **0.282** | **0.164** | **0.316** | **0.424** | **0.198** | 0.184 | **0.159** | **0.832** | 10s |

$$
\begin{cases}
\text{Diff}_{\text{policy}}^{v} = \parallel \mathbf{v}_\theta(\mathbf{x}_t^{v,w}, t) - \mathbf{v}^{v,w} \parallel_2^2 - \parallel \mathbf{v}_\theta(\mathbf{x}_t^{v,l}, t) - \mathbf{v}^{v,l} \parallel_2^2, \\
\text{Diff}_{\text{ref}}^{v} = \parallel \mathbf{v}_{\text{ref}}(\mathbf{x}_t^{v,w}, t) - \mathbf{v}^{v,w} \parallel_2^2 - \parallel \mathbf{v}_{\text{ref}}(\mathbf{x}_t^{v,l}, t) - \mathbf{v}^{v,l} \parallel_2^2, \\
\text{Diff}_{\text{policy}}^{a} = \parallel \mathbf{v}_\theta(\mathbf{x}_t^{a,w}, t) - \mathbf{v}^{a,w} \parallel_2^2 - \parallel \mathbf{v}_\theta(\mathbf{x}_t^{a,l}, t) - \mathbf{v}^{a,l} \parallel_2^2, \\
\text{Diff}_{\text{ref}}^{a} = \parallel \mathbf{v}_{\text{ref}}(\mathbf{x}_t^{a,w}, t) - \mathbf{v}^{a,w} \parallel_2^2 - \parallel \mathbf{v}_{\text{ref}}(\mathbf{x}_t^{a,l}, t) - \mathbf{v}^{a,l} \parallel_2^2, \\
\mathcal{L}_{\text{DPO}}^{av} = -\mathbb{E}_{t\sim[0,1],(\mathbf{x}_0^{a,w},\mathbf{x}_0^{v,w},\mathbf{x}_0^{a,l},\mathbf{x}_0^{v,l})\sim\mathcal{D}} \left[ \log \sigma\left(-\beta_v\left(\text{Diff}_{\text{policy}}^{v}-\text{Diff}_{\text{ref}}^{v}\right) - \beta_a\left(\text{Diff}_{\text{policy}}^{a}-\text{Diff}_{\text{ref}}^{a}\right)\right)\right]
\end{cases}
\tag{6}
$$

AV-DPO promotes well-aligned outputs while suppressing misaligned ones using curated preference pairs, with flow matching loss added for regularization (Hung et al., 2024) to avoid overfitting.

# 4 EXPERIMENTS

Sec. 4.1 and Sec. 4.2 present the setup and results of the main experiments, while Sec. 4.3 provides an in-depth analysis of the key modules. More ablation studies are included in Sec. D.

## 4.1 EXPERIMENTAL SETUP

**Implementation Details.** Our model is built upon Wan2.1-1.3B-T2V (Wan et al., 2025) and progressively adapted for joint audio-video generation: audio pre-training, audio-video SFT, and audio-video DPO. Rectified flow (Liu et al., 2023) is adopted as the noise scheduler for diffusion optimization. All the hyper-parameter details are provided in Sec. B.1.

**Training Datasets.** For audio data, we directly adopt the 780K audio-text pairs collected by JavisDiT (Liu et al., 2025c) for audio pre-training. For video data, we filtered 330K audio-video-text triplets from TAVGBench for audio-video SFT, along with an additional 25K samples for audio-video DPO. Further details and investigations are provided in Sec. B.2 and Sec. D.2.

**Evaluation Benchmarks.** We mainly follow JavisDiT (Liu et al., 2025c) to conduct experiments on JavisBench (10,140 samples for the main experiments) and JavisBench-mini (1,000 samples for ablation studies). The evaluation covers 11 metrics across various audio-video dimensions, including quality, consistency, and synchrony. Further details and explanations are provided in Sec. B.3.

## 4.2 MAIN RESULTS

As shown in Tab. 1, our JavisDiT++ model significantly outperforms previous JAVG methods in all dimensions of audio-video generation. In particular, we surpass UniVerse-1 (Wang et al., 2025a) by a large margin on quality and consistency, which also adopts Wan2.1-1.3B (Wan et al., 2025) as the backbone. This improvement is attributed to the unified and efficient design of MS-MoE, rather than simply stitching two pretrained models together as in UniVerse-1. On the other hand, our model also substantially outperforms both JavisDiT and UniVerse-1 on synchrony metrics, benefiting from

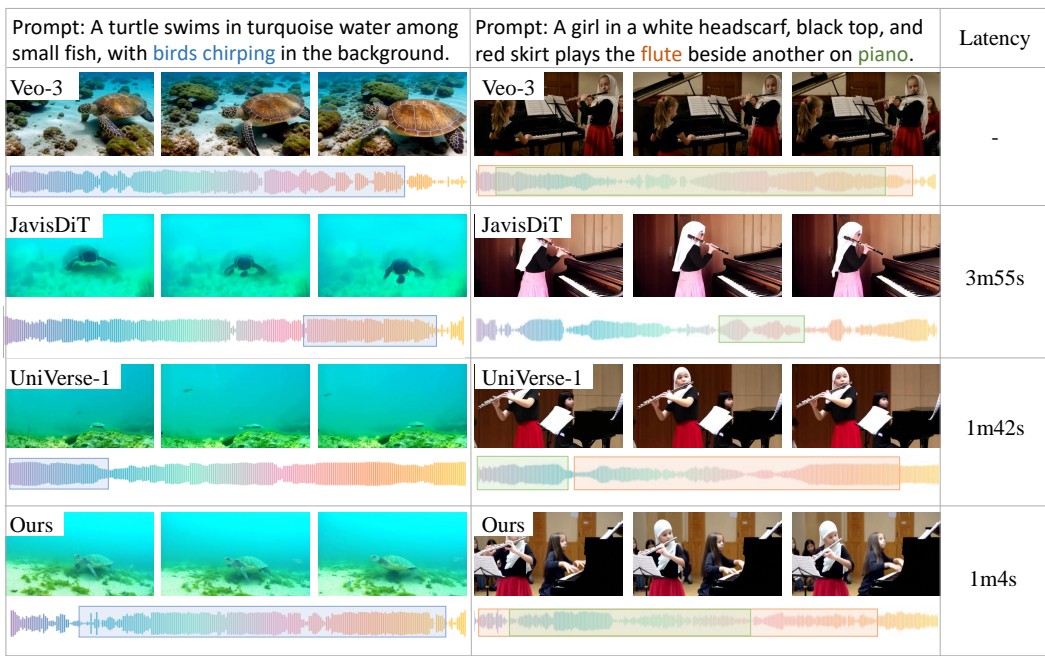

Figure 6: Generations from recent JAVG models. Best viewed in zoom or supplementary materials.

the explicitly designed TA-RoPE strategy that provides accurate audio-video temporal alignment. Fig. 6 further presents qualitative comparisons, where our generated results consistently surpass JavisDiT and UniVerse-1, narrowing the gap to the advanced Veo3. In addition, Fig. 6 also states that our model introduces only 1.6% additional inference cost over the Wan2.1 backbone (1m3s), significantly more efficient than two-stream methods like UniVerse-1 and JavisDiT.

## 4.3 IN-DEPTH ANALYSIS

In this section, we take JavisBench-mini (Liu et al., 2025c) with 1,000 prompts as the test set for evaluation. To avoid redundancy, we report 7 out of the 11 full metrics of audio-video generation.

Table 2: Investigation on architectural designs to adapt Wan2.1-T2V to joint audio-video generation.

| Arch Design | Quality | | Consistency | | | Synchrony | |
|---|---|---|---|---|---|---|---|
| | FVD ↓ | FAD ↓ | TV-IB ↑ | TA-IB ↑ | AV-IB ↑ | JavisScore ↑ | DeSync ↓ |
| Shared-DiT + LoRA | 227.6 | 6.51 | **0.283** | 0.138 | 0.127 | 0.098 | 0.934 |
| Shared-DiT + Full-FT | 269.3 | 5.66 | 0.276 | 0.159 | 0.164 | 0.137 | 0.945 |
| MS-MoE (Ours) | **221.3** | **5.51** | **0.283** | **0.163** | **0.194** | **0.153** | **0.807** |

**The proposed modality-specific MoE is capable of JAVG.** Tab. 2 compares three different strategies for performing audio pre-training and audio-video SFT within a unified model. First, we reuse Wan2.1-T2V as the shared DiT (Zhao et al., 2025) and apply either LoRA or full-parameter finetuning for text-to-audio-video (T2AV) adaptation, which serve as two important baselines. According to Tab. 2, the LoRA scheme suffers from poor audio quality and consistency due to its limited trainable capacity; the full-finetuning scheme, on the other hand, shifts too many parameters during the audio pre-training stage, which severely degrades video quality and consistency. In contrast, our MS-MoE design preserves strong video generation ability while equipping the model with high-quality audio generation, and simultaneously maintains excellent audio-video synchrony. In addition, Sec. D.2 reveals that ensuring good data quality is the foundation to increase the sample quantity to improve training efficacy, providing a new insight to scale up JAVG models in the future.

**Our LoRA configuration is a suitable choice.** Fig. 7 illustrates the effects of different settings for T2AV adaptation, where "A-LoRA" and "A-noLoRA" refer to whether adding LoRA during

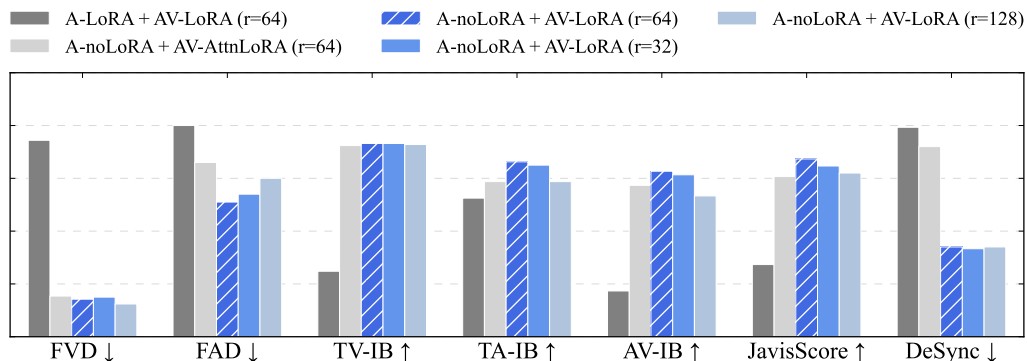

Figure 7: Comprehensive ablation studies on LoRA configurations.

audio pretraining, "AV-AttnLoRA" and "AV-LoRA" refer to adding LoRA to the attention blocks or to the whole DiT during audio-video joint training, and "r" denotes the compression rank. Fig. 7 reports normalized scores for each evaluation metric to compare the relative differences, where we can draw several conclusions: (1) Adding LoRA to the attention layers during the audio pre-training stage (A-LoRA) cannot improve audio generation quality but significantly reduces video generation performance, as it alters parameters in the video branch. (2) Compared with adding LoRA only to the attention layers (AV-AttnLoRA), also applying LoRA to FFN (AV-LoRA) leads to a notable improvement in audio-video generation performance, since the T2AV adaptation task remains relatively challenging. (3) The model performance is not highly sensitive to the choice of LoRA rank and alpha parameters; empirically, setting rank = 64 brings slightly better performance.

**The proposed TA-RoPE synchronization mechanism is effective and efficient.** Tab. 3 compares several designs for audio-video generation synchrony. First, although both the ST-Prior proposed in JavisDiT (Liu et al., 2025c) and the frame-level cross-attention used in UniVerse-1 (Wang et al., 2025a) improve synchrony, they bring an unaffordable increase in inference latency. In contrast, our

Table 3: Comparison of synchronization mechanisms.

| Mechanism | JavisScore ↑ | DeSync ↓ | Latency ↓ |
|---|---|---|---|
| None | 0.142 | 0.942 | **1m4s** |
| ST-Prior | 0.145 | 0.863 | 1m10s |
| FrameAttn | 0.124 | 0.850 | 1m22s |
| TA-RoPE (Ours) | **0.153** | **0.807** | **1m4s** |
| TA-RoPE + ST-Prior | 0.155 | 0.856 | 1m10s |
| TA-RoPE + FrameAttn | 0.151 | 0.802 | 1m22s |

TA-RoPE strategy achieves better performance with zero additional inference cost. On the other hand, TA-RoPE can also be combined with ST-Prior or FrameAttn to yield slightly better synchrony, but we ultimately discard these combinations to maintain inference efficiency and overall simplicity. Due to space limitations, a systematic investigation of TA-RoPE is placed at Sec. D.1.

**The AV-DPO design is reasonable and effective.** As Sec. D.3 validates the hyper-parameter (including $\beta$ and learning rate) choices, Tab. 4 mainly compares different reward strategies for selecting win–lose pairs in DPO training. First, applying modality-agnostic strategies like Average-Micro (averaging across all metrics before ranking (Liu et al., 2025b)) and Average-Macro (ranking within each metric and then averaging (Xue et al., 2025)) fails to achieve consistent improvements in audio-video generation. This is because they may form a win sample by combining better video but worse audio, which conflicts with eq. (6). In contrast, calculating rewards separately for audio and video and ensuring modality-consistent chosen pairs (*e.g.*, Modality-Micro/Macro) effectively improves generation quality, consistency, and synchrony. Meanwhile, removing normalization (*i.e.*, w/o norm) reduces the accuracy of pair ranking due to scale and range differences across rewards, which in turn degrades DPO performance. Likewise, discarding ground-truth samples and forming pairs only from generated candidates (*i.e.*, w/o gt) even gets worse results, as differences among generated samples are often too small to guide preference shifts.

## 4.4 HUMAN EVALUATION

In this section, we further conduct user studies to more comprehensively evaluate the performance of joint audio–video generation through human evaluation. Specifically, we reuse the 100 prompts

Table 4: Investigation on the effectiveness of different AV-DPO reward strategies.

| Reward Design | Quality | | Consistency | | | Synchrony | |
|---|---|---|---|---|---|---|---|
| | FVD ↓ | FAD ↓ | TV-IB ↑ | TA-IB ↑ | AV-IB ↑ | JavisScore ↑ | DeSync ↓ |
| None (baseline) | 221.3 | 5.51 | 0.283 | 0.163 | 0.194 | 0.153 | 0.807 |
| Average-Micro | 199.7 | 5.28 | 0.281 | 0.166 | 0.199 | 0.154 | 0.810 |
| Average-Macro | 203.3 | 5.31 | 0.281 | 0.166 | 0.196 | 0.152 | 0.810 |
| Modality-Micro | **198.5** | **5.32** | **0.284** | **0.168** | **0.201** | **0.156** | 0.776 |
| Modality-Macro | 201.1 | 5.41 | 0.282 | 0.166 | 0.197 | **0.156** | **0.773** |
| Modality-Micro (w/o norm) | 210.0 | 5.34 | 0.281 | 0.167 | 0.197 | 0.153 | 0.821 |
| Modality-Micro (w/o gt) | 234.7 | 5.43 | 0.281 | 0.164 | 0.197 | 0.154 | 0.833 |

from Fig. 2 and ask different models to generate corresponding audio–video outputs. For each prompt, we randomly sample the outputs of two models and recruit three volunteers to perform blind win–tie–lose preference judgments. The averaged scores across annotators are used as the final evaluation metric.

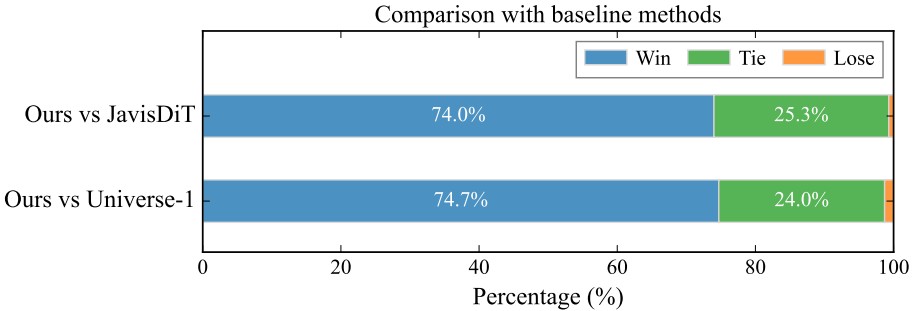

Figure 8: Subjective comparison on generation with baseline methods.

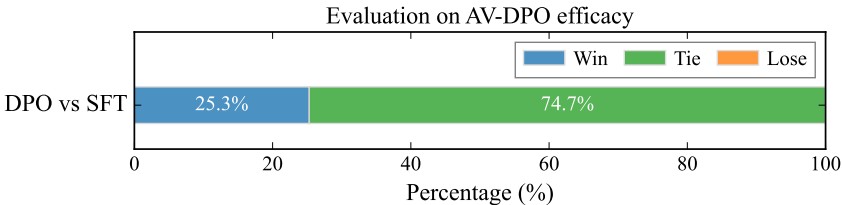

Figure 9: Subjective evaluation on the efficacy of the AV-DPO strategy.

**Our model substantially outperforms the baselines.** Fig. 8 shows that our method surpasses JavisDiT (Liu et al., 2025c) and UniVerse-1 (Wang et al., 2025a) by more than 70% in human preference. These subjective results align well with the objective evaluations in Tab. 1, jointly validating the effectiveness of our approach.

**The AV-DPO strategy effectively improves generation preference.** Although the gains from DPO were relatively modest in the previous objective evaluations, Fig. 9 demonstrates that the DPO-enhanced model produces over 25% more videos favored by human annotators, which further supports our motivation.

# 5 CONCLUSION

This work presents JavisDiT++, a concise and efficient framework for native joint audio-video generation. By introducing the MS-MoE design for modality-specific quality enhancement, the TA-RoPE strategy for explicit temporal alignment, and the AV-DPO algorithm for preference alignment, our model achieves state-of-the-art performance in quality, consistency, and synchrony. Built upon Wan2.1-1.3B-T2V and trained with only 1M data entries, our JavisDiT++ significantly outperforms existing open-source approaches while maintaining efficiency. We believe this work can establish an important milestone for JAVG and open new directions for future research in the field.

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

# A DISCUSSION

## A.1 POTENTIAL LIMITATIONS

While our framework demonstrates state-of-the-art performance in joint audio-video generation, several limitations remain and open promising directions for future work:

- **Training Data Scale**. Our model is trained on roughly 1M entries, which, although efficient, may constrain scalability compared with larger proprietary systems. Expanding the dataset with more diverse and high-quality audio-video pairs could further improve generalization and robustness.

- **Model Size**. We adopt a 1.3B-parameter backbone with parameter-efficient adaptations. Scaling up to larger backbones may unlock stronger representational capacity, especially for capturing subtle temporal and semantic correlations across modalities.

- **Full-Parameter Training**. Our approach relies on parameter-efficient tuning (e.g., LoRA). Exploring full-parameter finetuning could provide additional performance gains, albeit with higher computational cost.

- **Controllable Generation**. Current experiments focus on general text-to-audio-video generation. Extending controllability to domains such as music or speech, with fine-grained control over rhythm, pitch, timbre, or lexical content, is an important next step.

- **Unified Cross-Modal Generation**. Beyond text-conditioned JAVG, broader tasks such as audio-to-video (A2V), video-to-audio (V2A), and audio-image-to-video (AI2V) offer opportunities for a unified multimodal generative framework. Developing models that seamlessly perform across these modalities would mark a significant milestone toward general-purpose audio-visual content generation.

We hope these directions inspire future research toward building more scalable, controllable, and unified multimodal generative systems.

## A.2 ETHICS STATEMENT

All datasets and models used in this work are publicly available on the internet and do not involve any private or sensitive information. In addition, part of the DPO data we plan to release is generated by models themselves, ensuring that no personal privacy is infringed.

## A.3 REPRODUCIBILITY STATEMENT

We provide detailed descriptions of model design, training, and evaluation in both the main paper and the appendix. Furthermore, all code, pretrained checkpoints, and processed datasets will be publicly released to ensure full reproducibility of our results.

## A.4 LLM USAGE STATEMENT

Large Language Models (LLMs) were used solely as writing assistants, including tasks such as language polishing and presentation refinement. They were not involved in the conception of core ideas or designs.

# B DETAILED IMPLEMENTATIONS

## B.1 MODEL DETAILS

**Audio VAE**. We reuse and freeze the audio encoder and decoder from AudioLDM2 (Liu et al., 2024b). All 1D audio signals are resampled to 16 kHz and converted into 64-bin mel-spectrograms using a window size of 64 ms and a hop size of 10 ms. The spectrograms are then $8\times8$ compressed via VAE into 8-channel audio embeddings. To further reduce the token count, we apply a $2\times2$ patchify operation before feeding the tokens into the DiT for diffusion and denoising.

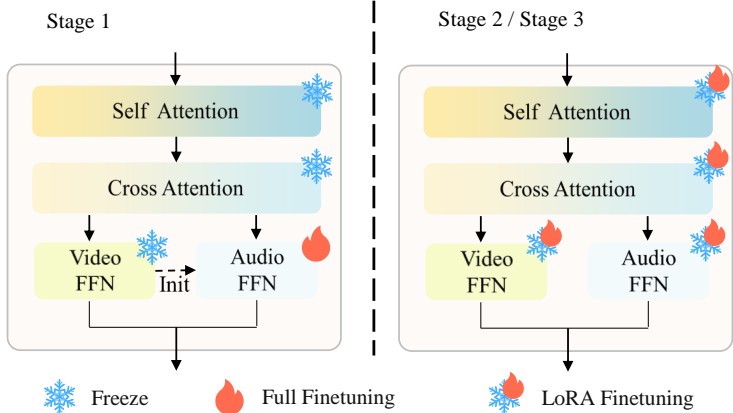

Figure A1: Illustration of trainable parameters at different stages.

Table A1: Detailed settings for the three-stage training pipeline.

| Setting | Stage 1 | Stage 2 | Stage 3 |
|---|---|---|---|
| training purpose | Audio PreTrain | Audio-Video SFT | Audio-Video DPO |
| trainable modules | Audio FFN/Embedder/Head | LoRA | LoRA |
| trainable params | 794M | 121M | 121M |
| learning rate | 1e-4 | 1e-4 | 1e-5 |
| warm-up steps | 1000 | 1000 | 100 |
| weight decay | 0.0 | 0.0 | 0.0 |
| training samples | 780K | 330K | 25K |
| resolution | - | dynamic | dynamic |
| duration | dynamic | dynamic | dynamic |
| batch size | dynamic | dynamic | dynamic |
| epoch | 50 | 2 | 1 |
| GPU days (H100) | 16 | 16 | 3 |

**Video VAE**. We reuse and freeze the video VAE from Wan2.1 (Wan et al., 2025). Except for the first frame, all subsequent video frames are temporally compressed by a factor of 4, while every frame is spatially compressed by $8\times8$, resulting in 16-channel video embeddings. To ensure further compactness, these video embeddings are also compressed with a $2\times2$ spatial patchify operation before being fed into the DiT.

**Text Encoder**. We also reuse and freeze Wan2.1's umT5-xxl (Chung et al., 2023) as the text-encoder, whose context length is 512.

**Backbone DiT**. Our model is built on the powerful Wan2.1-1.3B-T2V (Wan et al., 2025) model, and progressively extends from text-to-video generation to text-conditioned joint audio-video generation. The Wan2.1-1.3B base model has 30 layers with a hidden dimension of 1536. We keep the original parameters frozen throughout the entire training process, updating only the newly introduced audio FFN (along with audio embedder, audio head, etc) and the LoRA components at different stages. The final model has only 2.1B parameters after merging LoRA components. Detailed training settings are presented as follows.

## B.2 TRAINING DETAILS

**Details of the three-stage training pipeline.** Our model progressively extends from Wan2.1-1.3B-T2V (Wan et al., 2025) to JAVG through three stages: (1) Audio Pre-Training, where the Audio FFN is trained on 780K audio-text pairs for 50 epochs with a learning rate of 1e-4; (2) Audio-Video SFT, where LoRA is applied to train on 330K audio-video-text triplets for 1 epoch with a learning rate of 1e-4; and (3) Audio-Video DPO, where the LoRA parameters are retained and further trained

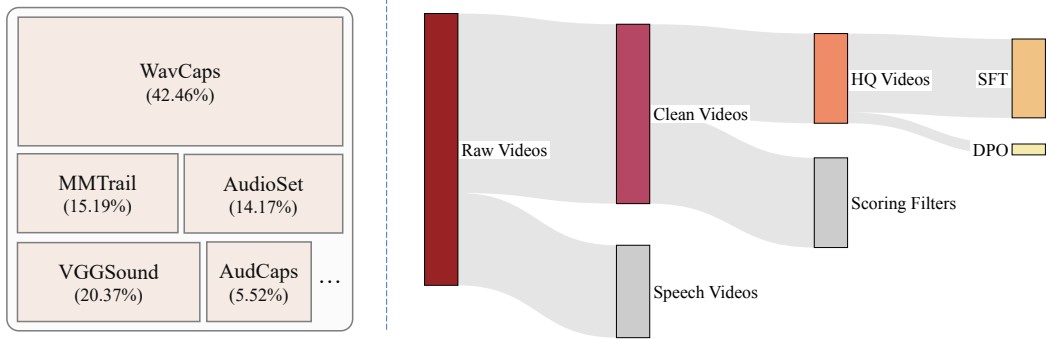

Figure A2: **(Left)**: Diversified audio-text sources. **(Right)**: Data filtering process from TAVGBench.

on 25K entries for 1 epoch with a learning rate of 1e-5. The specific trainable modules and other training settings can be found in Fig. A1 and Tab. A1.

**Details of the audio-video training data.** Fig. A2 demonstrates the detailed data composition and filtering procedure. For audio data, we directly adopt 780K audio-text pairs from Javis-DiT (Liu et al., 2025c) for audio pretraining, which covers various public datasets, including AudioSet (Gemmeke et al., 2017), AudioCaps (Kim et al., 2019), VGGSound (Chen et al., 2020), WavCaps (Mei et al., 2024), Clotho (Drossos et al., 2020), ESC50 (Piczak, 2015), GTZAN (Sturm, 2013), MACS (Martín-Morató & Mesaros, 2021), and UrbanSound8K (Salamon et al., 2014). We apply no data filtering strategy to ensure maximal text-to-audio generation capability spanning general sound, music, and speech.

For video data, we adopt a subset of 1.1 million text-video-audio triplets from TAVGBench (Mao et al., 2024) and conduct a series of filtering strategies. We first eliminate a large part of videos containing human speech by using the FunASR (Gao et al., 2023) detection tool, and then follow OpenSora (Zheng et al., 2024) to filter out the videos with relatively lower quality through aesthetic scoring (Schuhmann, 2022), flow (motion) scoring (Xu et al., 2023), and OCR scoring (Liao et al., 2020). In the final filtered pool, 330K data entries are divided for audio-video SFT, and the other 25K samples are used for audio-video DPO to avoid overlapping. We also provide empirical evaluations on the data diversity and quality in Sec. D.2.

### B.3 EVALUATION DETAILS

**Evaluation Setup.** We mainly follow JavisDiT (Liu et al., 2025c) to evaluate the models on JavisBench, which consists of 10,140 prompts for joint audio-video generation in various real-world scenarios. In ablation studies, we take the JavisBench-mini for fast evaluation, where the 1,000 prompts are randomly selected from the total 10,140 samples from JavisBench. All models are required to generate 240P, 4-second sounding videos for quantitative evaluation.

JavisBench provides a comprehensive evaluation of quality, consistency, and synchrony for audio-video generation results. Here, we briefly introduce the mechanisms of each evaluation dimension:

- **Audio / Video Quality**: measuring the perceptual quality of the generated audio and video, including (1) *Fréchet Video Distance (FVD)*: $\text{FVD} = \|\mu_r - \mu_g\|_2^2 + \text{Tr}(\Sigma_r + \Sigma_g - 2(\Sigma_r \Sigma_g)^{1/2})$, where $(\mu_r, \Sigma_r)$ and $(\mu_g, \Sigma_g)$ are the mean and covariance of ground-truth and generated video features extracted by a pretrained I3D encoder (Carreira & Zisserman, 2017). Lower is better, indicating the generated video distribution is closer to the real one; (2) *Kernel Video Distance (KVD)*: similar to FVD, but estimates distribution differences via a kernel-based method (Kernel Inception Distance style), which is more stable on smaller datasets; lower is better; and (3) *Fréchet Audio Distance (FAD)*: same concept as FVD, but computed on audio features extracted by a pretrained AudioClip model (Guzhov et al., 2022), measuring distribution distance between generated and real audio; lower is better.

- **Text Consistency**: evaluating how well the generated audio and video semantically match the input text description, including (1) *ImageBind (Girdhar et al., 2023) text-video cosine similarity*: $\text{sim}(t, v) = \frac{f_{\text{text}}(t) \cdot f_{\text{video}}(v)}{\|f_{\text{text}}(t)\| \cdot \|f_{\text{video}}(v)\|}$; (2) *ImageBind text-audio cosine similarity:* same pro-

cess but with the audio encoder $f_{\text{audio}}$; (3) *CLIP-Score*: using CLIP (Radford et al., 2021) to compute semantic similarity between text and video (video frames are sampled, encoded, and averaged); and (4) *CLAP-Score*: using CLAP (Wu* et al., 2023) to compute semantic similarity between text and audio.

- **Audio–Video Semantic Consistency**: measuring the semantic alignment between generated audio and generated video, including (1) *ImageBind audio-video cosine similarity*, encoding both modalities into the same space and computing cosine similarity between video and audio features; and (2) *Audio-Visual Harmony Score (AVHScore)*: introduced in TAVGBench (Mao et al., 2024) as a way to quantify how well the generated audio and video align semantically in a shared embedding space. It is defined by computing the cosine similarity between each video frame and the entire audio, then averaging across all frames: $\text{AVHScore} = \frac{1}{N}\sum_{i=1}^{N}\cos\big(f_{\text{frame}}(v_i),\ f_{\text{audio}}(a)\big)$. A higher AVHScore indicates stronger audio–video semantic consistency. Note that we remove the CAVP-Score (Luo et al., 2023) used in JavisDiT (Liu et al., 2025c) because this metric keeps a range from 0.798 to 0.801 and cannot capture the difference when evaluating semantic consistency.

- **Audio–Video Spatio-Temporal Synchrony**: evaluating spatiotemporal alignment in generated audio-video pairs, including (1) *JavisScore*: a new metric proposed in JavisDiT (Liu et al., 2025c). The core idea is to use a sliding window along the temporal axis to split the audio-video pair into short segments. For each segment, compute cross-modal similarity with ImageBind and take the mean score: $\text{JavisScore} = \frac{1}{N}\sum_{i=1}^{N}\sigma(a_i, v_i),\quad \sigma(v_i, a_i) = \frac{1}{k}\sum_{j=1}^{k}\underset{\min}{\text{top-}k}\{\cos\big(E_v(v_{i,j}), E_a(a_i)\big)\}$; and (2) *DeSync*: a metric adapted from Synchformer(Iashin et al., 2024), which measures fine-grained temporal misalignment between audio and video streams. Specifically, it estimates the temporal offset of audio–visual events by performing a 21-category classification task (predicting the offset/asynchrony degree ranging from -10 to 10 and taking the absolute values). A lower DeSync score indicates better synchronization.

**Compared Methods**. Since open-source models that support text-conditioned joint audio-video (JAVG) generation are still very limited, we follow JavisDiT (Liu et al., 2025c) and include cascaded generation pipelines (*i.e.*, T2A+A2V (Yariv et al., 2024a; Jeong et al., 2023) and T2V+V2A (Jeong et al., 2024; Xing et al., 2024; Zhang et al., 2024; Cheng et al., 2025)) for comparison. In particular, AudioLDM2 (Liu et al., 2024b) is adopted to conduct the preliminary T2A generation for subsequent T2A+A2V evaluation, while OpenSora-v1.2 (Zheng et al., 2024) is used to perform the preliminary T2V task for subsequent T2V+V2A methods. For JAVG-capable models, we treat the unconditional MMDiffusion (Ruan et al., 2023) as a simple baseline, while focusing on comparisons against text-conditional models like JavisDiT (Liu et al., 2025c) and UniVerse-1 (Wang et al., 2025a). For JavisDiT, we directly downloaded its released checkpoints and performed JavisBench generation locally, enabling us to compute the newly introduced DeSync (Iashin et al., 2024) metric and update other metrics consistently. For UniVerse-1, since it essentially supports audio-video generation from Text + Reference Image, we use the first frame of our model's generated video as the reference image, allowing UniVerse-1 to perform audio-synchronized image animation.

**Details of Fig. 2**. In Sec. 1, we present a radar chart (Fig. 2) to illustrate the performance differences among various JAVG models. Specifically, we randomly sample 100 prompts from JavisBench (Liu et al., 2025c) for audio-video generation. For open-source models, we run inference locally, while for Veo3 (DeepMind, 2025), we obtain results via its API. After collecting outputs from all models, we follow the DPO reward setup in Sec. 3.4 to comprehensively evaluate the performance, including (1) VideoAlign (Liu et al., 2025b) and AudioBox (Tjandra et al., 2025) to assess video and audio quality; (2) ImageBind (Girdhar et al., 2023) to compute semantic similarity across TV-Align, TA-Align, and AV-Align; and (3) SynchFormer (Iashin et al., 2024) to compute DeSync as the synchrony metric. Since DeSync is defined as a "lower-is-better" score, whereas all other metrics are "higher-is-better", we invert DeSync values in Fig. 2 to ensure visual consistency.

## C  DETAILED COMPARISON WITH RELATED WORKS

**Architectural Difference with Recent JAVG Models.** Fig. A3 presents a comparison between our model design and recent JAVG approaches. First, UniForm (Zhao et al., 2025) attempts to use

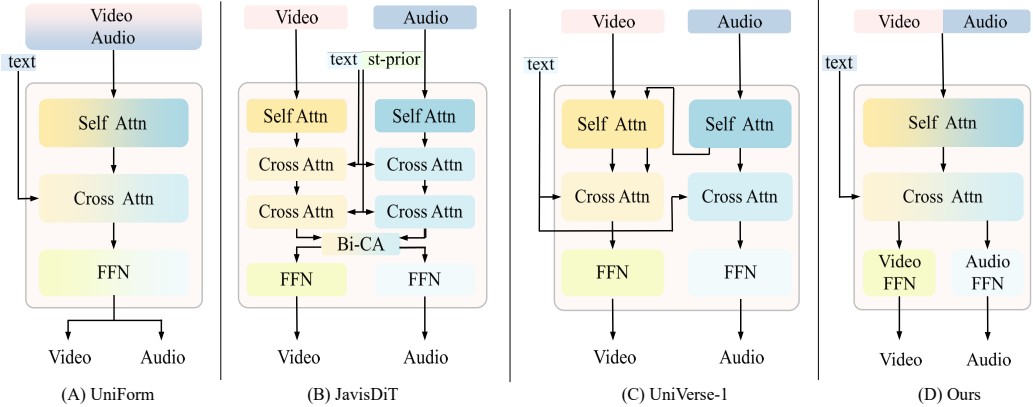

Figure A3: Architectural comparison with Uniform, JavisDiT, and UniVerse-1.

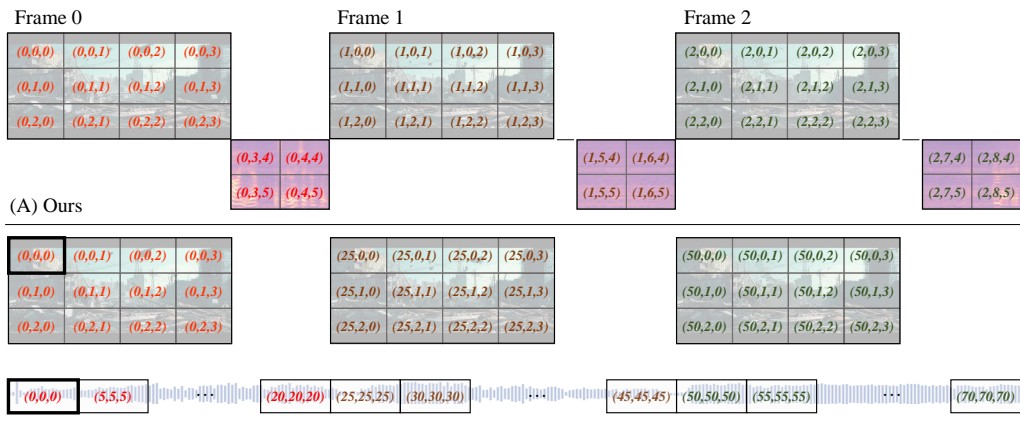

Figure A4: Comparison of our audio-video frame interleaving with Qwen2.5-Omni's strategy.

a single set of attention and FFN parameters to process both audio and video tokens. This poses a significant challenge when extending a pretrained T2V model with audio generation capability while preserving its original video generation performance, as validated in Tab. 2. To address this, JavisDiT (Liu et al., 2025c) introduces a dual-stream architecture with separate parameter sets for video and audio generation, along with ST-Prior and frame-level audio-video bidirectional cross-attention to enhance synchrony. However, this design introduces a large number of additional parameters, making training more difficult and inference more expensive. UniVerse-1 (Wang et al., 2025a), on the other hand, employs a pretrained T2V model (Wan et al., 2025) and a pretrained T2A model (Gong et al., 2025), with a complex cross-attention mechanism to enable information exchange between audio and video, which still suffers from inefficiencies in both training and inference stages.

In contrast to these prior works, our proposed framework is simpler and more effective: it uses shared attention to enable cross-modal interaction between audio and video tokens, while adopting modality-specific FFNs to enhance intra-modal modeling quality. This design achieves a better trade-off between efficiency and performance, and provides higher scalability. This idea is also explored by a concurrent work, JoVA (Huang et al., 2025b), which further disentangles the QKV projectors for audio and video tokens within the self-attention block.

**Design Difference with Qwen2.5-Omni's RoPE Strategy.** Considering audio-video alignment, our TA-RoPE design shares some similarity with Qwen2.5-Omni (Xu et al., 2025)'s RoPE solution: both approaches align audio and video through the position IDs along a specific dimension to ensure temporal alignment. However, as shown in Fig. A4, there exists a key difference between us: whether the position IDs of audio and video tokens overlap. Qwen2.5-Omni treats audio, like lan-

guage, as a 1D token sequence and assigns the same position IDs across the three RoPE dimensions. This inevitably introduces overlaps between audio and video, *e.g.*, an audio token with position ID $(0,0,0)$ coinciding with a video token $(0,0,0)$, or potentially $(25,25,25)$, $(50,50,50)$, and so on. For multimodal understanding models such as Qwen2.5-Omni, such overlaps have negligible influence on inference performance. However, for generative tasks like JAVG, overlaps cause non-trivial position confusion and lead to performance degradation, as quantitatively demonstrated in Sec. D.1.

By comparison, our TA-RoPE design treats audio as a 2D image (represented by the form of mel-spectrogram (Liu et al., 2024b)). When ensuring temporal alignment by matching the 0-th dimension of position IDs between audio and video along the time axis, we introduce adaptive offsets in the other two dimensions of the mel-spectrogram corresponding to video width and height (as formulated in eq. (4)). This strategy completely avoids overlap between the two modalities and enables more accurate audio-video synchrony.

**About AudioGen-Omni's RoPE Strategy**. Focusing on the domain of video-to-audio generation, AudioGen-Omni (Wang et al., 2025c) also identifies RoPE as a key component for achieving temporal alignment between audio and video, and proposes a specific design. However, due to the lack of detailed descriptions in their paper and the unavailability of their code, we are currently unable to conduct a thorough comparative analysis.

## D  ADDITIONAL EXPERIMENTS

### D.1  ABLATION ON POSITIONAL ENCODING STRATEGY

This section systematically investigates the impact of different audio position encoding strategies (illustrated in Fig. A5) on final audio-video generation quality, including:

- **Vanilla**, ignores video positions entirely, encoding audio purely along the time and frequency axes of its mel-spectrogram: $R_a(t,m) = (t,t,m)$
- **Interpolate**, aligns the audio temporal axis with video frames by interpolating intermediate IDs: $R_a(t,m) = \left(t \cdot \frac{T_v}{T_a}, t \cdot \frac{T_v}{T_a}, m\right)$
- **Interleave**, aligns the audio temporal axis (0-th dimension) with the video temporal axis, while unfolding the other two dimensions along the mel-spectrogram: $R_a(t,m) = \left(\left[t \cdot \frac{T_v}{T_a}\right], t, m\right)$
- **Interleave+Offset**, similarly aligns the audio 0-th dimension with the video temporal axis, but shifts the remaining two dimensions by the video width and height to fully avoid overlapping position IDs between modalities: $R_a(t,m) = \left(\left[t \cdot \frac{T_v}{T_a}\right], t+H, m+W\right)$ (eq. (4)).

We conduct full audio-pretraining and AV-SFT training on Wan2.1-1.3B-T2V (Wan et al., 2025) to examine how these strategies affect both audio and video generation. Results in Tab. A2 lead to three main conclusions:

1. Preserving integer audio position IDs is essential for audio quality — e.g., Interpolate performs significantly worse, since frozen attention layers during audio pretraining cannot learn relative offsets represented by fractional IDs;
2. The more overlap exists between audio and video position IDs, the poorer the video quality becomes (Vanilla → Interleave → Interleave+Offset), consistent with our analysis in Sec. C;
3. Generation models require non-overlapping position IDs to disentangle modalities, as well as temporal alignment along one axis for audio-video synchrony (*e.g.*, Vanilla yields much worse synchrony than Interleave-based designs).

Based on these findings, we adopt Interleave+Offset as our final position encoding scheme.

### D.2  INVESTIGATION ON TRAINING DATA QUALITY AND DIVERSITY

In this section, we conduct an in-depth investigation into the impact of diversity and quality of training data during the Audio-Video SFT stage. Specifically, we first construct a low-quality dataset

Figure A5: Illustration of different audio positional encoding strategies.

Table A2: Ablation study on positional encoding strategy for both audio and video generation.

| Strategy | AudioCaps | | | JavisBench-mini (audio) | | | JavisBench-mini (video) | | | | | |
|---|---|---|---|---|---|---|---|---|---|---|---|---|
| | FAD ↓ | TA-IB ↑ | CLAP ↑ | FAD ↓ | TA-IB ↑ | CLAP ↑ | FVD ↓ | TV-IB ↑ | CLIP ↑ | AV-IB ↑ | JavisScore ↑ | DeSync ↓ |
| AudioLDM2 | 5.06 | **0.200** | **0.460** | 8.81 | 0.153 | 0.360 | - | - | - | - | - | - |
| JavisDiT-audio | 5.19 | 0.164 | 0.356 | 8.11 | 0.152 | 0.381 | - | - | - | - | - | - |
| Vanilla | 6.03 | 0.193 | 0.411 | 6.41 | **0.157** | 0.417 | 238.2 | 0.281 | 0.316 | 0.184 | 0.142 | 0.918 |
| Interpolate | 12.87 | 0.155 | 0.325 | 10.73 | 0.149 | 0.349 | 239.9 | 0.282 | **0.320** | 0.183 | 0.144 | 0.912 |
| Interleave | 5.49 | 0.195 | 0.420 | **6.20** | 0.153 | **0.418** | 225.8 | **0.284** | 0.318 | 0.187 | 0.144 | 0.829 |
| Interleave+Offset | **4.65** | 0.198 | 0.420 | 6.81 | 0.154 | 0.417 | **221.3** | 0.283 | 0.317 | **0.200** | **0.153** | **0.807** |

of 720K text–audio–video triplets by extracting a large portion of speech videos (Gao et al., 2023) from the full data pool (see Sec. B.2). Next, we follow OpenSora (Zheng et al., 2024) to filter out low-quality data using multiple scoring metrics such as aesthetics (Schuhmann, 2022) (thresholding at 0.4), motion (Xu et al., 2023) (thresholding at 0.1), and OCR (Liao et al., 2020) (thresholding at 5.0), resulting in a medium-quality dataset of 330K samples, which is the set adopted for AV-SFT in the main paper. Finally, we further raise the aesthetic threshold from 0.4 to 0.45, obtaining a high-quality dataset of 120K samples for comparative analysis. After audio pretraining, we conduct experiments with different data compositions in the AV-SFT stage, and the results are presented in Tab. A3. Accordingly, several conclusions can be drawn:

First, data with high quality but low diversity cannot bring optimal performance. Models trained solely on the 120K high-quality dataset underperform those trained on the 330K medium-quality dataset. This is because transitioning from unimodal audio/video generation to joint generation is a relatively challenging task, requiring sufficient data quantity or diversity to enable the model to acquire new capabilities.

Second, data quality is also indispensable. Models trained on the 720K low-quality dataset show clearly inferior generation quality compared to those trained on the 330K medium-quality dataset. This degradation occurs because low-quality data undermine the priors learned by the Wan2.1 (Wan et al., 2025) backbone during pretraining on high-quality data. Even introducing a second round of SFT with 120K high-quality or 330K medium-quality data cannot fully recover the lost performance.

In summary, we adopt the 330K medium-quality dataset for AV-SFT training as a reasonable trade-off. We believe that further enhancing both the diversity and quality of training data will yield better scaling properties.

Table A3: Ablation study on training data composition in the AV-SFT stage.

| Data | Quality | Epoch | Quality | | Consistency | | | Synchrony | |
|---|---|---|---|---|---|---|---|---|---|
| | | | FVD ↓ | FAD ↓ | TV-IB ↑ | TA-IB ↑ | AV-IB ↑ | JavisScore ↑ | DeSync ↓ |
| 120K | High | 1.0 | 230.6 | 6.24 | 0.282 | 0.154 | 0.183 | 0.144 | 0.838 |
| 120K | High | 2.0 | 239.9 | 5.81 | **0.283** | 0.157 | 0.189 | 0.151 | 0.811 |
| 120K | High | 3.0 | 233.7 | 5.64 | 0.281 | 0.159 | 0.188 | 0.148 | 0.820 |
| 330K | Medium | 1.0 | 225.5 | 5.62 | **0.283** | 0.161 | 0.190 | 0.145 | 0.822 |
| 330K | Medium | 2.0 | 221.3 | 5.51 | **0.283** | **0.163** | **0.194** | **0.153** | **0.807** |
| 720K | Low | 0.5 | 229.8 | 5.53 | 0.281 | 0.154 | 0.182 | 0.142 | 0.830 |
| 720K | Low | 1.0 | 217.8 | 5.64 | 0.280 | 0.159 | 0.185 | 0.145 | 0.825 |
| +120K | Low+High | 2.0 | 223.8 | 5.46 | 0.282 | 0.161 | 0.191 | 0.150 | 0.823 |
| +330K | Low+Med | 1.0 | **212.0** | **5.45** | **0.283** | 0.160 | 0.187 | 0.146 | 0.824 |

### D.3 ABLATION ON HYPER-PARAMETERS OF AV-DPO

In this section, we further investigate the stability of hyperparameters in the AV-DPO algorithm proposed in eq. (6), focusing primarily on two factors: the choice of $\beta$ and the learning rate. Since different parameters influence the magnitude of the loss unevenly, we adopt *implicit accuracy* as a unified proxy metric to evaluate the impact of various hyperparameter settings:

$$Acc^a = \frac{1}{N} \sum_{i=1}^{n} \mathbb{I}(\text{Diff}_{\text{policy}}^a < \text{Diff}_{\text{ref}}^a); \quad Acc^v = \frac{1}{N} \sum_{i=1}^{n} \mathbb{I}(\text{Diff}_{\text{policy}}^v < \text{Diff}_{\text{ref}}^v) \quad (A1)$$

Recalling eq. (6), the implicit accuracy can measure whether the model successfully shifts toward the distribution of the chosen data while moving away from that of the rejected data. The experimental results, shown in Fig. A6 and Fig. A7, lead to the following conclusions:

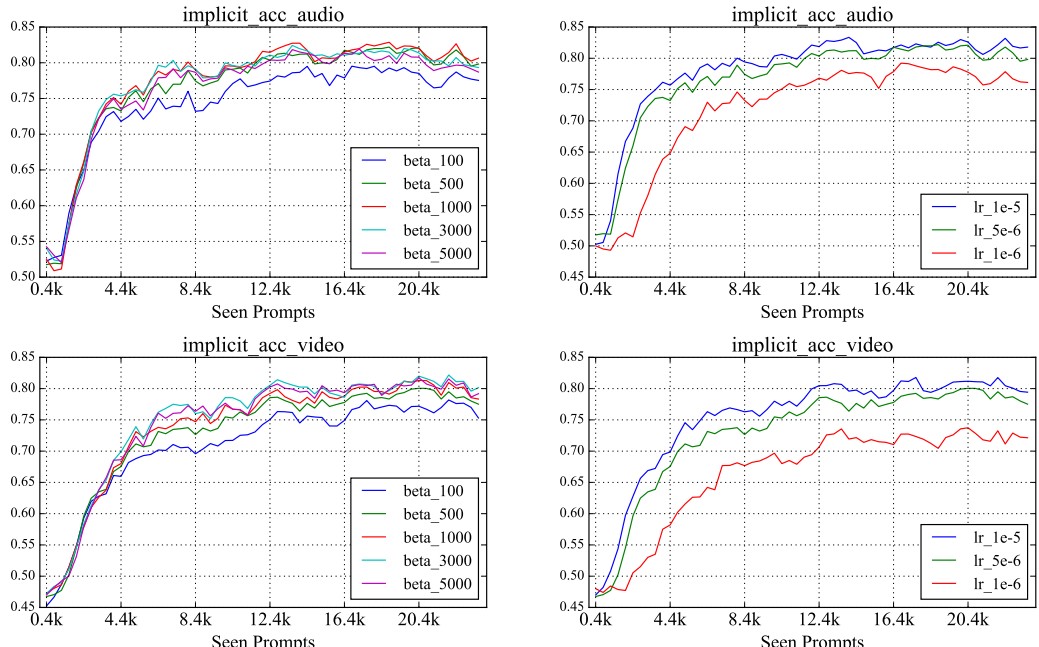

Figure A6: Implicit accuracy on $\beta$ selections.   Figure A7: Implicit accuracy on learning rates.

**Analysis on $\beta$ selection.** In the original DPO algorithm, $\beta$ primarily controls the divergence between the policy model and the reference model (Rafailov et al., 2023; Liu et al., 2025b). As shown in Fig. A6, audio achieves its best performance at $\beta = 1000$, whereas video converges faster and

reaches higher final accuracy at $\beta = 3000$. This observation is closely related to our proposed framework. Our model is built upon the pretrained Wan2.1-T2V, which already aligns well with human preferences; thus, a larger $\beta$ (*e.g.*, 3000 or 5000) is needed to keep the video policy model closer to the reference model. In contrast, the audio branch is newly trained and initially less aligned with human preferences, so a smaller $\beta$ (*e.g.*, 1000) is required to shift the model closer to the preferred data distribution. However, setting $\beta$ too small (*e.g.*, 100) leads to excessive divergence from the reference model and overfitting to imperfect preference data. Based on these findings, we set $\beta = 3000$ for the audio DPO loss and $\beta = 1000$ for the video DPO loss to achieve relatively better performance.

**Analysis on learning rate.** Fig. A7 presents three experimental groups with learning rates set to $1 \times 10^{-5}$, $5 \times 10^{-6}$, and $1 \times 10^{-6}$, respectively. The results show that $1 \times 10^{-5}$ achieves both the fastest convergence and the highest final accuracy. This observation is consistent with prior works (Liu et al., 2025b), which suggests that setting the learning rate in the DPO stage to approximately one-tenth of that in the SFT stage is a suitable choice. Therefore, we adopt a learning rate of $1 \times 10^{-5}$ for training.

## D.4 MORE VISUALIZATIONS

Fig. A8 and Fig. A9 showcase additional cases of joint audio-video generation, illustrating the strong generative capability of our proposed model across multiple dimensions.

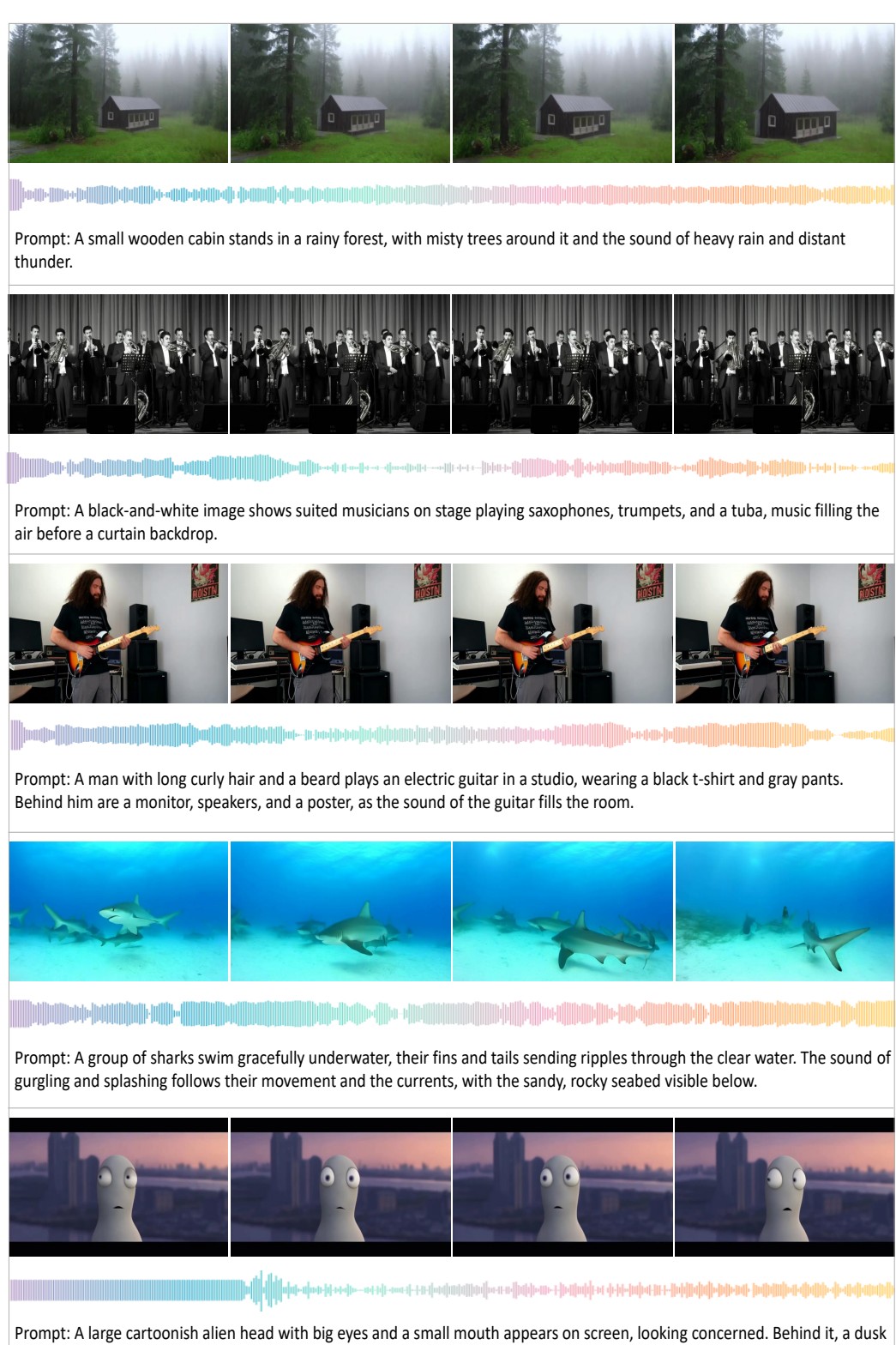

Figure A8: More examples for high-quality audio-video generation results.

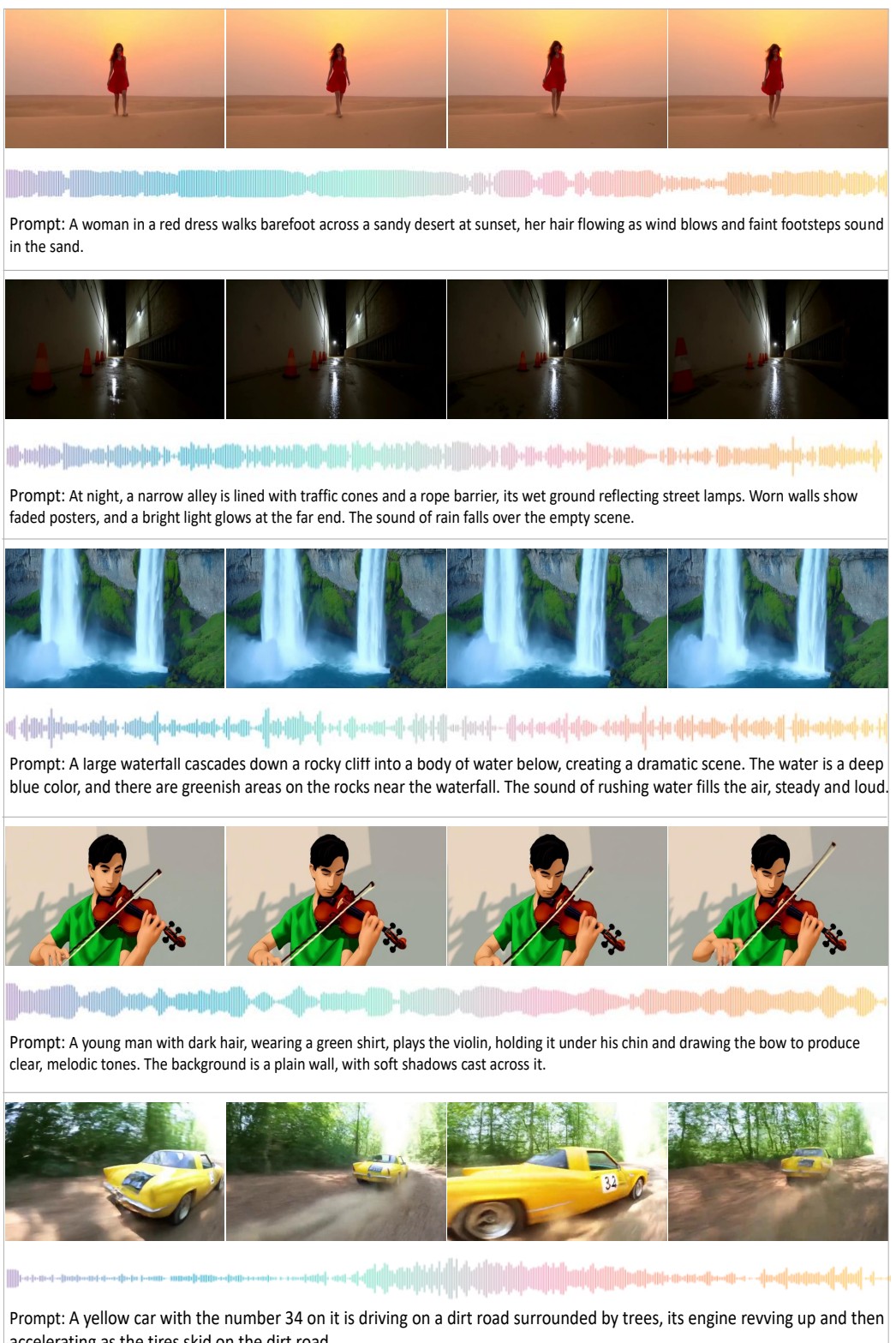

Figure A9: More examples for high-quality audio-video generation results.

