# OpenReview forum: "JavisDiT++: Unified Modeling and Optimization for Joint Audio-Video Generation"
_ICLR.cc/2026/Conference — ICLR 2026 Poster_

### Official Review · Reviewer_Zh7Q · 2025-10-27

**Soundness:** 3
**Presentation:** 2
**Contribution:** 3
**Rating:** 6
**Confidence:** 5

**Summary:**

The paper presents a new framework for audio-video joint generation, which features three new techniques. First, a modality-specific mixture-of-experts (MS-MoE) design is adopted, which leads to both cross-modal alignment and single-modality generation quality. Second, a temporal-aligned RoPE (TA-RoPE) strategy is introduced for facilitating frame-level synchronization between audio and video tokens. Third, an audio-video direct preference optimization (AV-DPO) method is applied to align model outputs with human preference across quality, consistency, and synchrony dimensions. Experimental results with a model built on Wan2.1-1.3B demonstrate that the proposed framework outperforms previous approaches both qualitatively and quantitatively. Ablation studies justify the design of the proposed modules.

**Strengths:**

1. Although there is room for improvement in presentation, the paper itself is written well enough to make readers understand the authors' motivation, the proposed method, and the experimental results.
2. The proposed framework is well-designed. The ablation studies are comprehensive and support the design.
3. The experimental results are good both quantitatively and qualitatively.

**Weaknesses:**

Although the proposed framework is well-designed and works better than the previous ones (JavisDiT and UniVerse-1), there is one concern/question. Newly introduced FFNs for audio tokens will modulate audio tokens so that the attention layers trained for video tokens can deal with audio tokens as well. This idea is great. However, as for the first Transformer block, audio tokens (the output of the audio encoder) will be fed into the self-attention and cross-attention layers as they are. It is suspected that the self-attention and cross-attention layers in the first Transformer block will not be able to deal with audio tokens very well. So, inserting another FFN between the audio encoder and the first Transformer block and training it in the pretraining phase might help the first self-attention and cross-attention layers process audio tokens.

**Questions:**

I have a question/concern, which I provided in "Weaknesses". I would appreciate it if the authors could share their thoughts on it. If the authors' response is convincing, I will raise my rating.

---

> ### Comment · Reviewer_Zh7Q · 2025-10-27
> **Minor comments**
>
> Let me list minor issues about presentation
>
> - L.82: "show the cap" -> "show**s** the **g**ap"
> - L.97: use **\citep{}** here, instead of \cite{} or \citet{}
> - Figure 7 and Lines 432-439:
>
> It is a little hard for readers to understand what "A-LoRA", "A-noLoRA", "AV-LoRA", and "AV-AttnLoRA" mean. Readers will need to consider it carefully. Using labels in the explanation as follows would help readers.
> > (1) Adding LoRA to the attention layers during the audio pre-training stage (**A-LoRA**) cannot improve audio generation quality but significantly reduces video generation performance, as it alters parameters in the video branch. (2) Compared with adding LoRA only to the attention layers (**AV-AttnLoRA**), also applying LoRA to FFN (**AV-LoRA**) leads to a notable improvement in audio-video generation performance, since the T2AV adaptation task remains relatively challenging.
>
> - Table 4 and Lines 453-464:
>
> It is hard for readers to understand what "Average-Micro/Macro" and "Modality-Micro/Macro" mean. Using labels as follows would help readers.
> > First, applying modality-agnostic micro averaging (**Average-Micro**; averaging across all metrics before ranking (Liu et al., 2025b)) or macro averaging (**Average-Macro**; ranking within each metric and then averaging (Xue et al., 2025)) fails to achieve consistent improvements in audio-video generation. This is because such strategies may form a win sample by combining better video but worse audio, which conflicts with eq. (5). In contrast, calculating rewards separately for audio and video and ensuring modality-consistent chosen samples (**Modality-Micro**) effectively improves generation quality, consistency, and synchrony. Meanwhile, removing normalization for single-dimension rewards (i.e., w/o norm) reduces the accuracy of pair ranking due to scale and range differences across rewards, which in turn degrades DPO performance. Likewise, discarding ground-truth samples and forming pairs only from generated candidates (i.e., w/o gt) even gets worse results, as differences among generated samples are often too small to guide preference shifts.
>
> - Figure 3:
>
> Placing the "TA-RoPE" block between the video/audio encoders and the Transformer block is confusing. To my understanding, RoPE is applied inside the self-attention layers of Transformer blocks. What is done between the video/audio encoders and the Transformer block would be just to flatten and interleave video and audio tokens.

---

> ### Author Response · Authors · 2025-11-23
> **Response to Reviewer Zh7Q**
>
> We thank the reviewer for the valuable time and effort to evaluate our paper. We hope our detailed responses below can help address the reviewer's concerns.
>
> > W1/Q1: As for the first Transformer block, audio tokens (the output of the audio encoder) will be fed into the self-attention and cross-attention layers as they are. Inserting another FFN between the audio encoder and the first Transformer block and training it in the pretraining phase might help the first self-attention and cross-attention layers process audio tokens.
>
> **Ans**:
> Thank you for pointing this out. In fact, after passing through the VAE encoder and before entering the DiT, we also include a learnable audio embedder layer to bridge the two feature spaces. For simplicity, this component was not shown in Figure 3, and we have now added a clarification in the caption.
>
> > Minor 1: About presentation:
> (1) L.82: "show the cap" -> "shows the gap"
> (2) L.97: use \citep{} here, instead of \cite{} or \citet{}
> (3) Figure 7 and Lines 432-439: Using labels would help readability.
> (4) Table 4 and Lines 453-464: Using labels would help readability.
>
>
> **Ans**:
> Thank you for pointing this out. We have corrected it in the revised version.
>
> > Minor 2: For Figure 3,
> Placing the "TA-RoPE" block between the video/audio encoders and the Transformer block is confusing. RoPE is applied inside the self-attention layers of Transformer blocks. What is done between the video/audio encoders and the Transformer block would be just to flatten and interleave video and audio tokens.
>
> **Ans**:
> Yes, RoPE is indeed applied within the attention layers of each block. However, as discussed at the end of Section 3.3, base models with a built-in RoPE mechanism (e.g., Wan) can directly perform TA-RoPE inside the attention module, whereas models without RoPE (e.g., OpenSora) must instead interleave the tokens physically beforehand. Ultimately, both approaches enable the DiT to learn and perceive the temporal alignment between video and audio tokens.

---

> ### Comment · Reviewer_Zh7Q · 2025-11-24
>
> Thank you for your work. I went through all the reviewers' comments and your responses. I appreciate all of them. Almost all of my concerns have been addressed, and I am considering raising my rating; however, one concern remains unresolved, specifically regarding Minor 2.
>
> To my understanding, TA-RoPE consists of two processes: (1) audio-visual temporal alignment and (2) rotary matrix multiplication. The second process is called PoPE, and it was originally proposed in [Su et al. (2024)](https://www.sciencedirect.com/science/article/abs/pii/S0925231223011864), as you cite in your paper. In general, the rotary matrix multiplication is applied to query and key vectors based on their relative positions, but not applied to value vectors. This process is implemented inside an attention layer in both [Wan](https://github.com/Wan-Video/Wan2.1/blob/7c81b2f27defa56c7e627a4b6717c8f2292eee58/wan/modules/model.py#L149) and [OpenSora](https://github.com/hpcaitech/Open-Sora/blob/d0cd5ac50da79e9a9d2285a952d4dcd806e6c8fc/opensora/models/mmdit/math.py#L22) implementations.
>
> I understand that the first one (audio-visual temporal alignment) can be achieved by manipulating position IDs as you do. However, many readers will think TA-RoPE includes both two processes, and I still believe that placing the "TA-RoPE" block between the video/audio encoders and the DiT block is confusing. I recommend clarifying that what is done before entering the DiT is only audio-visual temporal alignment in the figure, its caption, and/or the main text. If my understanding is incorrect, do not hesitate to point it out. I just want to make your paper less confusing, and I am open to discussion.

---

> > ### Author Response · Authors · 2025-11-24
> >
> > We truly appreciate the constructive feedback. After in-depth consideration and visualization attempts, we agree with the reviewer’s suggestion that placing TA-RoPE inside the attention layer indeed reduces conceptual complexity and helps avoid potential misinterpretation. We have updated Figure 3 accordingly in the revised manuscript to improve the overall clarity of the presentation.

---

> > > ### Comment · Reviewer_Zh7Q · 2025-11-24
> > >
> > > Thank you for listening to my comments. I recommend this paper. I have raised my rating.
> > >
> > > Let me keep monitoring discussions with the other reviewers. Thank you.

---

> > > > ### Author Response · Authors · 2025-11-24
> > > >
> > > > Thanks for your positive feedback and for raising your rating. Your comments genuinely helped us strengthen and clarify the paper. We sincerely appreciate your support.

---

### Official Review · Reviewer_uNjt · 2025-10-27

**Soundness:** 3
**Presentation:** 2
**Contribution:** 3
**Rating:** 6
**Confidence:** 4

**Summary:**

This paper presents a new method for efficiently constructing a model capable of jointly generating audio and video. Given a pretrained DiT-based video generation model, the proposed method first adds audio-specific FFNs to each DiT block, and these added FFNs are trained with text-audio pairs for text-to-audio generation. Then, the entire model is fine-tuned in a parameter-efficient manner using text-audio-video triplets for joint audio-video generation. Finally, the model is further fine-tuned by direct preference optimization with a dedicated audio-video preference dataset. Experimental results show that the model obtained through the proposed method achieves superior performance compared to existing models.

**Strengths:**

- The proposed method introduces a novel approach for efficiently constructing joint audio-video generation models. While many prior works use a separate audio branch in addition to a video branch, the proposed model only adds dedicated FFNs for audio.
- The proposed positional encoding is carefully designed to leverage pretrained video generation models while inducing strong audio-visual alignment via temporal encoding.
- This work explores direct preference optimization for joint audio-visual generation. The preference data is automatically constructed using several reward models. This approach is simple and reasonable, and could serve as a good baseline for future research.

**Weaknesses:**

- While the idea of applying DPO to joint generation models is interesting, its effect appears to be relatively small compared to the other proposed modifications. It would be beneficial to investigate the potential causes of this result.
  - For example, if the quality of the preference data is a factor, this could be clarified by a subjective evaluation of a small subset of the preference data. If the amount of preference data is insufficient, showing performance as a function of the amount of preference data used would be helpful.
- Some visualizations could be further improved:
  - Showing spectrograms rather than waveforms would be more informative for illustrating the semantics of the audio content.
  - What does the vertical axis in Figure 7 represent?

**Questions:**

- Why are several numbers (especially DeSync in sequential generation baselines) missing from Table 1? These are necessary for a comprehensive comparison of joint generation performance.
- What models are used for T2A (or T2V) in the sequential generation baselines?

---

> ### Author Response · Authors · 2025-11-23
> **Response to Reviewer uNjt**
>
> We thank the reviewer for the valuable time and effort to evaluate our paper. We hope our detailed responses below can help address the reviewer's concerns.
>
> > W1: The efficacy of applying DPO appears to be relatively small. It would be beneficial to investigate the potential causes of this result, e.g., the quality of the preference data or the amount of preference data.
>
> **Ans**:
> Thank you for the suggestion.
> - First, the primary factor is likely the quality of the preference data. From our manual inspection of 100 sampled preference pairs, we estimate an accuracy of 84%, and further improving preference quality would likely yield larger gains.
> - Second, data quantity is probably not the main limitation, as Figures A6 and A7 in Appendix A3 show that training has already reached a stable convergence with the current 25K samples. Without improving preference quality, simply increasing the amount of training data would have limited effect.
> - Finally, diversity may also be an issue. The generated video cases under the same prompt are often very similar, suggesting that expanding the pool of candidates may help enhance the diversity of preference data.
>
> > W2.1: Some visualizations could be further improved: Showing spectrograms rather than waveforms would be more informative for illustrating the semantics of the audio content.
>
> **Ans**:
> Thank you for the suggestion. Due to limited time, we were not able to revise the manuscript in this round. However, we will further examine the differences between waveform and spectrogram visualizations during the upcoming discussion period.
>
> > W2.2 What does the vertical axis in Figure 7 represent?
>
> **Ans**:
> The vertical axis in Figure 7 represents the normalized relative performance, and we have added this clarification in the revised Section 4.3. Since the score ranges differ across metrics, the corresponding normalization coefficients also vary. For this reason, we do not directly annotate the normalized scores in Figure 7, so as to avoid introducing additional cognitive burden for readers.
>
> > Q1: Why are several numbers (especially DeSync in sequential generation baselines) missing from Table 1? These are necessary for a comprehensive comparison of joint generation performance.
>
> **Ans**:
> Thank you for the suggestion. We reimplemented the generation process in JavisDiT to obtain the sequential-baseline outputs required for measuring DeSync. The updated results have been included in Table 1 of the revised manuscript. Notably, the T2V + A2V baseline now achieves a level of synchrony comparable to that of the joint audio–video generation models.
>
> > Q2: What models are used for T2A (or T2V) in the sequential generation baselines?
>
> **Ans**:
> We follow the setup of JavisDiT and use AudioLDM2[1] as the base model for T2A and Open-Sora[2] as the base model for T2V. The corresponding clarification has been added to Appendix B.3 of the revised manuscript.
>
> ---
>
> [1] Haohe Liu, Yi Yuan, et al. Audioldm 2: Learning holistic audio generation with self-supervised pretraining. IEEE TASLP, 2024.
>
> [2] Zangwei Zheng, Xiangyu Peng, et al. Open-sora: Democratizing efficient video production for all, 2024. URL https://github.com/hpcaitech/Open-Sora.

---

> > ### Comment · Reviewer_uNjt · 2025-11-25
> >
> > Thanks for the clarification. I do not have any further questions on my end.
> >
> > Although the reported accuracy of the preference annotation suggests there is room for improvement in making DPO more effective, the novelty of the model architecture and the empirical performance demonstrated in the paper are significant. I am inclined to accept this paper, although it is not a strong recommendation. I look forward to seeing the opinions of the other reviewers during the discussion phase.

---

> > > ### Author Response · Authors · 2025-11-26
> > >
> > > Thanks for the constructive comments to strengthen and clarify our work. We sincerely appreciate your support.

---

### Official Review · Reviewer_qjFx · 2025-10-30

**Soundness:** 2
**Presentation:** 2
**Contribution:** 2
**Rating:** 4
**Confidence:** 4

**Summary:**

This paper proposes a joint audio-video generation (JAVG) model built on the pretrained text-to-video model Wan2.1.
To enable synchronized audio and video generation, three mechanisms are introduced: a model design called MS-MOE for joint generation, a new RoPE variant (TA-RoPE) that manages both cross-modal interaction and single-modal structure, and an audio-video direct preference optimization method (AV-DPO) that aligns the model with human preference.
Experimental results demonstrate that the proposed method outperforms existing baselines.

**Strengths:**

- The proposed model design (MS-MOE with LoRA finetuning) is simple yet effective, achieving both a high cross-modal alignment and a high single-modal generation quality. It is also computationally efficient, as inference cost per token remains constant while the parameter size increases.
- The proposed TA-RoPE requires minimal engineering efforts. It provides a natural extension of Wan's RoPE to support both inter- and intra-modal interactions of audio and video.
- AV-DPO sounds novel, as applying DPO for aligning JAVG models with human preference is underexplored.

**Weaknesses:**

The primary concern lies in the **fairness and clarity of the experimental evaluation**.

For Table 1:
- It is unclear which T2A models are used for T2A+A2V and which T2V models are used for T2A+V2A. Also, many results of T2A+A2V and T2V+V2A baselines are missing.
- The authors re-evaluate JavisDiT, but the reported scores deviate substantially from those in the original paper. For instance, TA-IB = 0.197 (original) vs. 0.151 (this paper), CLIP = 0.325 vs. 0.308, AV-IB = 0.201 vs. 0.197, AVHScore =  0.183 vs. 0.179, and JavisScore = 0.158 vs. 0.154. These metrics consistently decrease, and in some cases, the original scores are higher than those of the proposed method.
- Only the final model (finetuned using AV-DPO) is reported. It would also be important to show results after the Audio-Video SFT stage to isolate the contributions of the model architecture and AV-DPO optimization.

For Fig.2:
- All evaluation metrics appear identical to those used in AV-DPO training, relying on the same reward models. Since only the proposed model is optimized for these metrics, this evaluation is biased and not directly comparable to other baselines.

For Fig.7:
- The definitions of A-LoRA, A-noLoRA, AV-AttnLoRA, and AV-LoRA are missing, making the figure difficult to interpret. A more detailed explanation is needed.


**Lack of justification for the TA-RoPE design.**
While the reviewer agrees that TA-RoPE is a reasonable engineering extension, the motivation behind the specific positional correspondence between audio and video modalities is not fully explained.
In particular, the mapping of video height and audio time, and video width and audio frequency, may introduce implicit correlations that lack perceptual grounding.
A brief visualization or analysis of the positional-similarity structure would help clarify how the TA-RoPE affects the interaction between audio and video modalities.


**Lack of subjective evaluation.**
Although the paper claims that AV-DPO improves perceptual alignment with human preference, no user study is conducted to support this claim.
Including even a small-scale user study would make the evaluation more convincing.

**Questions:**

- Did the authors evaluate the model performance after the Audio-Video SFT stage, prior to AV-DPO finetuning?
- For TA-RoPE, could the authors visualize or analyze the similarity structure between RoPE embeddings used for audio and video?
- For the training data used in AV-DPO, how did the authors sort the videos based on three scores?
- Is AV-DPO applicable to other existing JAVG models?
- Did the authors conduct a user study to compare the proposed method with baselines?

---

> ### Author Response · Authors · 2025-11-23
> **Response to Reviewer qjFx - Part I**
>
> We thank the reviewer for the valuable time and effort to evaluate our paper. We hope our detailed responses below can help address the reviewer's concerns.
>
> > W1.1: For Table 1, it is unclear which T2A models are used for T2A+A2V and which T2V models are used for T2A+A2V. Also, many results of T2A+A2V and T2V+V2A baselines are missing.
>
> **Ans**: We follow the setup of JavisDiT and use AudioLDM2 as the base model for T2A and Open-Sora as the base model for T2V. The corresponding clarification has been added to Appendix B.3.
> For pipeline generation methods, we also follow JavisDiT in omitting the standalone performance of the base models and focusing instead on the final audio–video pair quality. Therefore, the results of the base models themselves are not included.
>
> > W1.2: For Table 1, The authors re-evaluate JavisDiT, but the reported scores deviate substantially from those in the original paper. These metrics consistently decrease, and in some cases, the original scores are higher than those of the proposed method.
>
> **Ans**:
> First, since we needed to evaluate the DeSync results, we reproduced the generation outputs of JavisDiT and, in the process, re-reported the other metrics as well. Note that not all metrics decreased; for example, TV-IB increased from 0.151 (original) to 0.195 (ours). A likely reason is that the released evaluation code of JavisDiT exhibits slight inconsistencies with the performance reported in their paper.
> Ultimately, we ensure fairness by evaluating both JavisDiT and our model using the exact same evaluation pipeline and codebase in the uploaded anonymous repo  https://anonymous.4open.science/r/iclr26-19605-code/.
>
> > W1.3: For Table 1, only the final model (finetuned using AV-DPO) is reported. It would also be important to show results after the Audio-Video SFT stage to isolate the contributions of the model architecture and AV-DPO optimization.
>
> **Ans**:
> Thank you for the suggestion. In fact, in Table 4 of our manuscript, we have  already compare the improvements of different AV-DPO strategies over the SFT stage (the “baseline” row) on the 1,000 data points. Here, we include the performance of the AV-SFT model on the full JavisBench in the revised Table 1.
>
> | Model | AV-Quality   |  | Text-Consistency |  |  |  | AV-Consistency |  | AV-Synchrony |  |
> |---|---|---|---|---|---|---|---|---|---|---|
> |  | FVD ↓ | FAD ↓ | TV-IB ↑ | TA-IB ↑ | CLIP ↑ | CLAP ↑ | AV-IB ↑ | AVHScore ↑ | JavisScore ↑ | DeSync ↓ |
> | AV-SFT | 154.1  | 5.6 | 0.281 | 0.162 | 0.315 | 0.419 | 0.189 | 0.172 | 0.151 | 0.859 |
> | +AV-DPO | 141.5 | 5.5 | 0.282 | 0.164 | 0.316 | 0.424 | 0.198 | 0.184 | 0.159 | 0.832 |
>
> The results are consistent with the observations in Table 4:
> - AV-DPO provides marginal gains in text consistency, while yielding clear improvements in AV quality (e.g., FVD decreases from 154.1 to 141.5). This may be related to the common training strategy of freezing the text encoder while fine-tuning the DiT.
> - AV-DPO significantly enhances semantic consistency and spatiotemporal synchrony in the generated audio–video pairs, which aligns with our motivation: ensuring cross-modal preference consistency is essential for DPO to effectively improve joint audio–video generation quality.
>
> > W1.4: For Fig.2, All evaluation metrics appear identical to those used in AV-DPO training, relying on the same reward models. Since only the proposed model is optimized for these metrics, this evaluation is biased and not directly comparable to other baselines.
>
> **Ans**:
> To eliminate the potential bias in Figure 2, we additionally report the SFT-stage results below. As shown, even without AV-DPO, our model already outperforms the baseline models across all aspects.
>
> | Model | V-Quality | A-Quality | TA-Align | TV-Align | AV-Align | AV-Sync |
> |---|---|---|---|---|---|---|
> | JavisDiT | 1.02 | 4.28 | 2.61 | 1.31 | 1.80 | 0.75 |
> | UniVerse | 1.52 | 4.09 | 2.90 | 1.11 | 0.97 | 0.80 |
> | Ours-SFT | 2.43 | 4.84 | 2.97 | 1.64 | 1.87 | 0.94 |
>
> In fact, the purpose of Figure 2 is to evaluate model quality along six representative and computationally convenient dimensions, which are **independent** of the specific reward model used during DPO training. The results in the table further support our belief that, starting from the SFT model, using alternative reward models would also lead to consistent improvements across these six dimensions.
>
> > W1.5: For Fig.7, The definitions of A-LoRA, A-noLoRA, AV-AttnLoRA, and AV-LoRA are missing, making the figure difficult to interpret. A more detailed explanation is needed.
>
> **Ans**:
> Thank you for the suggestion. We have added the corresponding explanation in Section 4.3:
>
> _“A-LoRA” and “A-noLoRA” refer to whether adding LoRA during audio pretraining, “AV-AttnLoRA” and “AV-LoRA” refer to adding LoRA to the attention blocks or to the whole DiT during audio-video joint training, and “r” denotes the compression rank._

---

> ### Author Response · Authors · 2025-11-23
> **Response to Reviewer qjFx - Part II**
>
> > W2: Lack of justification for the TA-RoPE design. A brief visualization or analysis of the positional-similarity structure would help clarify how the TA-RoPE affects the interaction between audio and video modalities.
>
> **Ans**:
> We have provided a detailed discussion on the design of position IDs for audio and video tokens in Appendix C/D1, as well as in Figure A4/A5 and Table A2 of the manuscript. We include extensive visualizations and experiments to analyze their impact on the audio–video modalities. Our findings indicate that the two modalities must not share overlapping position IDs, and that applying a video-shaped offset to the audio token positions is an effective and efficient solution. This design preserves the absolute temporal alignment across modalities, maintains the intrinsic spatiotemporal structure within each modality, and avoids cross-modal interference.
>
> > W3: Lack of subjective evaluation. Although the paper claims that AV-DPO improves perceptual alignment with human preference, no user study is conducted to support this claim. Including even a small-scale user study would make the evaluation more convincing.
>
> **Ans**:
> Thank you for the suggestion. We selected 100 prompts from Figure 2 and asked both the SFT and DPO models to generate the corresponding sounding videos. Three volunteers were then recruited to conduct blind win–tie–lose preference evaluations, and the averaged results are shown below:
>
> | Comparison | DPO-win | DPO-tie | DPO-lose |
> |---|---|---|---|
> | DPO v.s. SFT | 25.3% | 74.7% | 0% |
>
> Accordingly, although the improvement brought by DPO appears relatively limited in the objective evaluations reported in the manuscript, the human study indicates that DPO indeed yields a 25.3% increase in human preference, which further supports our motivation.
>
> > Q1: Did the authors evaluate the model performance after the Audio-Video SFT stage, prior to AV-DPO finetuning?
>
> **Ans**:
> We provide the ablation results on JavisBench-mini in Table 4 of the manuscript, and the full-scale experiments as well as the human evaluation results have been added above.
>
> > Q2: For TA-RoPE, could the authors visualize or analyze the similarity structure between RoPE embeddings used for audio and video?
>
> **Ans**:
> Please see our response above.
>
> > Q3: For the training data used in AV-DPO, how did the authors sort the videos based on three scores?
>
> **Ans**:
> For each synthetic audio–video sample ((a, v)), we first compute scores along the three dimensions—audio, video, and audio–video—denoted as ((S^a, S^v, S^{a,v})). After ranking the candidates under the same prompt, we select a winning sample ((a^w, v^w)) whose scores in all three dimensions are higher than those of the losing sample ((a^l, v^l)); that is,
> $S^a(a^w) > S^a(a^l), S^v(v^w) > S^v(v^l), S^{a,v}(a^w, v^w) > S^{a,v}(a^l, v^l).$
> A formal description of this procedure is provided in Equation (5) of the revised manuscript.
>
>
> > Q4: Is AV-DPO applicable to other existing JAVG models?
>
> **Ans**:
> Sure! Our proposed AV-DPO is a model-agnostic and general algorithm that can be applied to any other JAVG model. We will continue to explore and integrate it with more advanced models in future work.
>
> > Q5: Did the authors conduct a user study to compare the proposed method with baselines?
>
> **Ans**:
> Similarly, we reused the 100 prompts from Figure 2 and collected the corresponding videos generated by different models. Three volunteers were then recruited to conduct blind win–tie–lose preference evaluations, and the averaged results are shown below:
>
>
> | Comparison | Ours-win | Ours-tie | Ours-lose |
> |---|---|---|---|
> | Ours v.s. JavisDiT | 74.0% | 25.3% | 0.7% |
> | Ours v.s. UniVerse-1 | 74.7% | 24.0% | 1.3% |
>
> These results confirm that our model also substantially outperforms the baseline models in human subjective evaluation, further validating the effectiveness of our approach.

---

> ### Comment · Reviewer_qjFx · 2025-11-25
>
> Thank you for your work and the detailed explanation.
>
> Most of my concerns have been addressed: the explanation of the experimental setups is clear, and the human study is convincing. Overall, the proposed module is well-designed (simple yet achieving strong empirical results), and the novelty of applying DPO to JAVG is evident. Given these points, I am currently inclined to accept.
>
> One follow-up question concerns the TA-RoPE design.
> In the proposed TA-RoPE, successive positional indices are assigned across the spatial dimensions of both video and audio. If I understand correctly, RoPE applies a rotation to the features using $\cos(m\theta_i)$ and $\sin(m\theta_i)$, where $m$ is the positional index. This suggests that audio patches in the upper-left regions are more similar to nearby video features than those in the lower-right regions. For example, in Fig. A5, the patch at (0,3,4) may exhibit greater similarity with the video patch at (0,2,3) than with the one at (0,4,5).
> Although this correlation naturally diminishes as the temporal offset increases, could the authors elaborate on this behavior?
> Similarly, to mitigate such biased correlations, one possible solution might be to use a larger offset rather than increments of 1. Have the authors considered this extension?

---

> > ### Author Response · Authors · 2025-11-26
> >
> > Thank you for the positive feedback, and we are glad that our previous revisions addressed the earlier concerns. Regarding the new question on TA-RoPE, we provide our reasoning and explanation as follows:
> >
> > 1. RoPE should be interpreted through relative positional relationships/angles rather than similarity.
> >    Taking the audio patch at positions (0,3,4) as an anchor, both the video patch at (0,2,3) and the audio patch at (0,4,5) have the same relative displacement (0,1,1), but in opposite directions. It is exactly this opposite direction that produces completely different position embeddings (the cosine channels remain unchanged, while the sine channels flip sign, causing a large change in the embedding). Therefore, as long as no positional ID overlap exists between modalities, as claimed in the paper, there will be no modality confusion.
> >
> > 2. We did consider using a very large offset, but doing so could cause the relative positional angles between audio and video tokens to exceed the maximum positional delta observed during video pretraining, thereby introducing an out-of-distribution issue. Moreover, choosing such a large offset would be highly ad hoc, reducing both interpretability and transferability.
> >
> > In summary, we believe that the current design remains a clean and reasonable solution.

---

> > > ### Comment · Reviewer_qjFx · 2025-11-27
> > >
> > > Thank you for the detailed explanation. My concerns have been satisfactorily addressed, and I am pleased to raise my score.

---

### Official Review · Reviewer_9Rrw · 2025-11-01

**Soundness:** 3
**Presentation:** 3
**Contribution:** 2
**Rating:** 6
**Confidence:** 4

**Summary:**

This paper proposes a new method for joint audio-video generation. In contrast to most prior works that model the two modalities as two separate streams, this paper proposes to use a mixture-of-experts design to allow frequent communication and joint modeling of both modalities. This paper also proposes to be a temporally aligned rope encoding scheme and a DPO method for preference alignment. The proposed approach achieves state-of-the-art performance.

**Strengths:**

- The joint modeling of audio and video tokens with separate FFN layers is simple and elegant, although these ideas already exist in prior works (e.g., BAGEL) under different contexts.
- The presented model has a strong empirical performance, setting a new state-of-the-art for the challenging problem of joint audio-video generation.
- The study of different finetuning strategies (e.g., Lora with different ranks) is interesting. It provides additional context and insights to the readers.
- The ablation study on data composition in the appendix is also informative.

**Weaknesses:**

- The proposed network seems to be significantly larger than the networks used in prior works. Can the authors also include the parameter counts and runtime in Table 1 for a more comprehensive comparison?
- The gains in Table 4 from using DPO seem to be very limited, except in FVD. How important is DPO? Are there any user studies or qualitative examples that support the use of DPO?
- There are no details on training data filtering. Section D2 contains a high-level sketch, but it does not mention which algorithms or what thresholds were used.
- The last update date of the linked repo is after the ICLR deadline.

**Questions:**

- Will the code and model be open-sourced?
- Will the filtered training dataset be released? This would be a significant contribution.

---

> ### Author Response · Authors · 2025-11-23
> **Response to Reviewer 9Rrw**
>
> We thank the reviewer for the valuable time and effort to evaluate our paper. We hope our detailed responses below can help address the reviewer's concerns.
>
> > W1: The proposed network seems to be significantly larger than the networks used in prior works. Can the authors also include the parameter counts and runtime in Table 1 for a more comprehensive comparison?
>
> **Ans**: Thank you for the suggestion. In the revised manuscript, we have added the parameter counts and runtimes of the relevant methods. For clarity, we also provide a comparison of the two core approaches below:
>
> | Model | #Params | Latency |
> |---|---|---|
> | JavisDiT | 3.1B | 30s |
> | UniVerse-1 | 6.5B | 13s |
> | Ours | **2.1B** | **10s** |
>
> Similar to prior work such as JavisDiT and UniVerse-1, our system also builds upon a ~1.3B T2V backbone to extend it to T2AV generation. With our efficiently designed MS-MoE architecture, we introduce only a minimal number of additional parameters (from 1.3B to 2.1B) while achieving the fastest runtime (10s) among the compared methods, further demonstrating the effectiveness and efficiency of our approach.
>
> > W2: The gains in Table 4 from using DPO seem to be very limited, except in FVD. How important is DPO? Are there any user studies or qualitative examples that support the use of DPO?
>
> **Ans**: Thank you for the suggestion. We selected 100 prompts from Figure 2 and asked both the SFT and DPO models to generate the corresponding sounding videos. Three volunteers were then recruited to conduct blind win–tie–lose preference evaluations, and the averaged results are shown below:
>
> | Comparison | DPO-win | DPO-tie | DPO-lose |
> |---|---|---|---|
> | DPO v.s. SFT | 25.3% | 74.7% | 0% |
>
> Accordingly, although the improvement brought by DPO appears relatively limited in the objective evaluations reported in the manuscript, the human study indicates that DPO indeed yields a 25.3% increase in human preference, which further supports our motivation.
>
> > W3: There are no details on training data filtering. Section D2 contains a high-level sketch, but it does not mention which algorithms or what thresholds were used.
>
> **Ans**: Thank you for pointing this out. Our data filtering pipeline follows the procedure described in Section B2. Specifically:
>
> - We first apply FunASR[1] to the 1.1M raw collected videos to remove samples containing speech, resulting in a 720K low-quality dataset.
>
> - We then apply aesthetic scoring[2] (threshold = 0.4), motion scoring[3] (threshold = 0.1), and OCR scoring[4] (threshold = 5.0), obtaining a 330K medium-quality dataset.
>
> - Finally, we increase the aesthetic scoring threshold to 0.45 to obtain a 120K high-quality dataset.
>
> Descriptions of these filtering tools and the corresponding threshold choices have been added to Section D2 of the revised manuscript.
>
>
> > Q1: Will the code and model be open-sourced?
>
> **Ans**: Of course. We will release the model and code as soon as possible to support progress in this research area.
>
> > Q2: Will the filtered training dataset be released? This would be a significant contribution.
>
> **Ans**: Regarding the training data, our dataset is constructed from TAVGBench[5] and thus faces similar YouTube-related copyright constraints. Therefore, we are currently unable to directly release all video data. We will first provide data-acquisition scripts, and will release the full dataset once the copyright issues are resolved.
>
> In addition, the synthesized DPO training data does not involve copyright restrictions and will be released ahead.
>
> ---
>
> [1] Zhifu Gao, Zerui Li, et al. Funasr: A fundamental end-to-end speech
> recognition toolkit. In INTERSPEECH, 2023.
>
> [2] Christoph Schuhmann. Improved aesthetic predictor, March 2022. URL https://github.com/christophschuhmann/improved-aesthetic-predictor.
>
> [3] Haofei Xu, Jing Zhang, et al. Unifying flow, stereo and depth estimation. IEEE TPAMI, 2023.
>
> [4] Minghui Liao, Zhaoyi Wan, et al. Real-time scene text detection
> with differentiable binarization. In AAAI, 2020.
>
> [5] Yuxin Mao, Xuyang Shen, et al. Tavgbench: Benchmarking text to audible-video generation. In ACM MM, 2024.

---

> > ### Comment · Reviewer_9Rrw · 2025-11-26
> >
> > I thank the authors for their response. It has addressed my concerns well. The two newly added tables are helpful to highlight the contributions of this work. I don't think there are any other major weaknesses, so I am raising my score to an 8.

---

> > > ### Author Response · Authors · 2025-11-26
> > >
> > > We truly thank the reviewer for the constructive comments and positive feedback, which help us to make our work much stronger. We sincerely appreciate your support.

---

### Author Response · Authors · 2025-12-01
**Rebuttal Summary**

Dear ACs, SACs, and PCs,

We thank the reviewers for their time and constructive discussion, and also feel sorry for the impact caused by the recent OpenReview bug. Below, we provide a concise summary of the reviewers’ initial concerns and the outcomes of the rebuttal and discussion phase (where scores had already improved from 6,4,6,6 → 8,6,6,8 before the large-scale leakage). We hope this summary helps the AC’s effort in making the final decision.


### Initial Concerns

The reviewers’ initial concerns were mainly focused on the following:

1. **Lack of human evaluation**:
Reviewers 9Rrw and qjFx suggested incorporating human evaluations to further validate the effectiveness of DPO. Reviewer qjFx additionally recommended including subjective comparisons against baseline methods.
2. **Need for additional factual evidence**:
Reviewer qjFx raised concerns regarding the full-scale evaluation of AV-DPO and fairness in comparisons with baselines, while reviewer 9Rrw requested additional comparisons in terms of parameter counts and runtime.
3. **Improving paper presentation**:
The reviewers provided suggestions from different angles, including improvements to data-processing descriptions, experimental result presentation, figure clarity, and other details.

### Rebuttal and Discussions

During the rebuttal period, we provided extensive experiments and detailed explanations that successfully addressed all concerns raised by the reviewers. Before the large-scale leakage on Nov. 27, the reviewers had already expressed clear positive feedback and increased their scores:

- Reviewer 9Rrw: “I don't think there are any other major weaknesses, so I am raising my score to an 8.”
**Rating raised 6 → 8, ~24 hours before the bug.**
- Reviewer qjFx: “My concerns have been satisfactorily addressed, and I am pleased to raise my score.”
**Rating raised 4 → 6, ~12 hours before the bug.**
- Reviewer uNjt: “I am inclined to accept this paper, although it is not a strong recommendation.”
**Rating remained 6, ~2 days before the bug.**
- Reviewer Zh7Q: “I recommend this paper. I have raised my rating.”
**Rating raised 6 → 8, ~3 days before the bug.**

### Conclusion

After substantial discussion, **the reviewers unanimously agreed that our work is simple and elegant, shows strong empirical performance, and provides additional context and insights, with all concerns resolved**. The scores improved accordingly to 8,6,6,8 prior to the large-scale leakage. We have incorporated all new experiments, discussions, and clarifications into the revised manuscript.

We once again thank the reviewers and AC for their efforts, and we hope this summary is helpful for the final decision.

Sincerely,

Authors

---

### Meta-Review · Area_Chair_3Jao · 2026-01-06

**Summary:**

Reviewers initially questioned the model's efficiency, the actual impact of DPO without human studies, and missing details on baseline comparisons and TA-RoPE.

**Reviewer Concerns:**

The authors addressed latency benchmarks, provided a human preference study, and clarified TA-RoPE’s logic and positioning. Re-evaluated baselines and filtering thresholds were included. No major technical concerns remain outstanding.

**Reviewer Scores:**

Reviewers 9Rrw, and Zh7Q raised their scores to 8,  qjFx raised to 6. Reviewer uNjt (6) is satisfied with the architecture and performance results. The final consensus is strongly positive.

---

### Decision · Program_Chairs · 2026-01-26

Accept (Poster)